# On Designing Diffusion Autoencoders for Efficient Generation and Representation Learning

## Abstract

Diffusion autoencoders (DAs) are variants of diffusion generative models that use an input-dependent latent variable to capture representations alongside the diffusion process. These representations can be used for tasks such as downstream classification, controllable generation, and interpolation. However, the generative performance of DAs relies heavily on how well the prior distribution over the latent variables can be modelled and subsequently sampled from. Better generative modelling is also the goal of another class of diffusion models—those that learn their forward (noising) process. While effective at adjusting the noise process in an input-dependent manner, they must satisfy additional constraints derived from the terminal conditions of the diffusion process. Here, we draw a connection between these two classes of models and show that certain design decisions (latent variable choice, conditioning method, etc.) in the DA framework—leading to a model we term DMZ—enable effective representations as evaluated on downstream tasks, including domain transfer, as well as more efficient modelling and generation with fewer denoising steps compared to standard diffusion models.

## 1 Introduction

Learning effective and efficient deep generative models with latent variables has been an open problem in machine learning for some time (Kingma & Welling, 2014; Bengio et al., 2014). Solving this problem would require satisfying three aims: generating data that matches the observed data distribution well, capturing relevant information in the latent variables that facilitates interventions or downstream use, and doing both in a computationally efficient manner.

Diffusion models (DMs) are a powerful class of deep generative models that excel at generation, with deep hierarchical models (e.g., DDPMs (Ho et al., 2020)) and score-based models (Song et al., 2021b) serving as foundations. However, these models are computationally expensive and are not set up to capture effective latent representations. Approaches to address efficiency have largely focussed on making generation faster and more robust (Song et al., 2021a; Nichol & Dhariwal, 2021; Ning et al., 2023; Okhotin et al., 2023). Approaches to capturing representations have largely focused on extracting them from pre-trained models, whether through latent codes (Zhang et al., 2022), internal activations (Yang & Wang, 2023; Xiang et al., 2023), analysis of degradation patterns (Yue et al., 2024), or by aiming to disentangle interpretable structures (Yang et al., 2023; Zhang et al., 2023a).

While most DMs assume a fixed forward noising process and focus on learning the reverse (denoising) process, recent work has explored additionally learning the forward (noising) process itself, leading to more efficient learning and better models (Kingma et al., 2021; Bartosh et al., 2024; Nielsen et al., 2024). Independent of this, a recently-developed variant of DMs called diffusion autoencoders (DAs) incorporates input-dependent latent variables to capture representations alongside the diffusion process to enable reconstruction, controllable generation and interpolations (Preechakul et al., 2022; Wang et al., 2023a). Their effectiveness at capturing such information, the ability to subsequently generate data well, and to potentially do so with fewer denoising steps all depend strongly on how well the latent variable is fit and can be sampled from during inference.

Here, we draw a connection between DMs that learn their forward process for better and more efficient models and DAs that capture latent representations. We show that certain design decisions with the DA framework, including the choice and dimensionality of latent variable, method of conditioning

the denoising process, and setup for the learning the latent distribution allow us to obtain the best of both worlds—simply defining this class of models DMZ; i.e., diffusion models with $z$. We show DMZ can learn effective representations as evaluated on downstream tasks, including a novel domain transfer setting for DAs, as well as more efficient learning, modelling, and generation with fewer denoising steps compared to standard DMs. Our contributions are:

1. to show that DAs with judicious modelling choices can be efficient diffusion models to both train and infer with,
2. to explore and ablate design decisions that enable such efficiency, and
3. to evaluate the quality of representations in such models, through their latent variable, on a variety of tasks.

## 2 BACKGROUND AND RELATED WORK

Diffusion models (DMs) gradually corrupt data into noise through a forward process and learn to reverse this corruption. Denoising Diffusion Probabilistic Models (DDPMs; Ho et al., 2020) established this foundational setup with a Markovian noising process.

Given sample $x_0$ from the data distribution $q(x_0)$ and a predefined noise schedule $(\beta_1, \ldots, \beta_T)$, the forward process simulates a Markov chain starting from data $x_0 \sim q(x_0)$, iteratively adding Gaussian noise over $T$ diffusion steps until obtaining a completely noisy image $x_T \sim \mathcal{N}(0, \mathbf{I})$:

$$q(x_{1:T} \mid x_0) = \prod_{t=1}^{T} q(x_t \mid x_{t-1}), \qquad q(x_t \mid x_{t-1}) = \mathcal{N}\left(x_t; \sqrt{1-\beta_t} x_{t-1}, \beta_t \mathbf{I}\right). \tag{1}$$

Given observation $x_0$, the noised sample at $t$ is derived as $x_t = \sqrt{\bar{\alpha}_t}\, x_0 + \sqrt{1-\bar{\alpha}_t}\, \epsilon$, where $\epsilon \sim \mathcal{N}(0, \mathbf{I})$, $\alpha_i = 1 - \beta_i$, and $\bar{\alpha}_t = \prod_{i=1}^{t} \alpha_i$. The reverse (denoising) process is parametrised by $\theta$:

$$p_\theta(x_{t-1}|x_t) = \mathcal{N}\left(x_{t-1}; \mu_\theta(x_t, t), \sigma_t^2 \mathbf{I}\right), \quad \sigma_t = \sqrt{\frac{1-\bar{\alpha}_{t-1}}{1-\bar{\alpha}_t} \beta_t}. \tag{2}$$

Instead of directly predicting the mean of the forward process posterior $\mu_\theta(x_t, t)$, Ho et al. (2020) propose training a neural network $\epsilon_\theta(\cdot)$ to predict the noise vector $\epsilon$ by optimising:

$$L(\theta) = \mathbb{E}_{x_0 \sim q(x_0),\, \epsilon \sim \mathcal{N}(0, \mathbf{I}),\, t \sim \mathcal{U}(\{1, \ldots, T\})} \left[\|\epsilon - \epsilon_\theta(x_t, t)\|^2\right]. \tag{3}$$

The noise predictor $\epsilon_\theta$ minimising (3) can be expressed in terms of the score of the marginal distribution of $x_t$, establishing a connection between denoising autoencoders and score-based modelling (Song et al., 2021b).

For inference, the reverse process is defined as $p_\theta(x_{t-1}|x_t) = \mathcal{N}\left(x_{t-1}; \mu_\theta(x_t, t), \sigma_t^2 \mathbf{I}\right)$, where

$$\mu_\theta(x_t, t) = \frac{1}{\sqrt{\alpha_t}}\left(x_t - \frac{1-\alpha_t}{\sqrt{1-\bar{\alpha}_t}} \epsilon_\theta(x_t, t)\right) \quad \text{and} \quad \sigma_t^2 = \frac{1-\alpha_{t-1}}{1-\alpha_t} \beta_t. \tag{4}$$

Markovian DMs were later extended to non-Markovian variants (Song et al., 2021a; Okhotin et al., 2023), where the input $x_0$ influences the denoising process, resulting in fewer steps required for inference. These models assume a fixed forward process and focus solely on learning the reverse denoising process.

**Diffusion models with learned forward process:** Recent work explores parametrising and learning the forward process (noising) as well as the denoising process. VDMs (Kingma et al., 2021), NFDMs (Bartosh et al., 2024) and DiffEnc (Nielsen et al., 2024) learn both the forward process $q_\phi(x_t|x_0, t)$ and the reverse process $p_\theta(x_0|x_t)$, and have been shown to achieve better log-likelihood, potentially requiring fewer steps for inference. Other work explores conditional diffusion and use of data-dependent priors (Lee et al., 2022) or shifts (Zhang et al., 2023b). This direction parallels the motivations behind hierarchical variational autoencoders (VAEs) (Vahdat & Kautz, 2020; Kuzina & Tomczak, 2024), which introduce multi-level latent structures to better capture data distributions.

**Diffusion autoencoders (DAs):** This class of models introduces a latent variable that guides denoising, enabling tasks such as retrieval and editing through learned representations (Preechakul et al., 2022; Wang et al., 2023a; Pandey et al., 2022; Hudson et al., 2024). Note that DAs are also effectively non-Markovian DMs as they condition denoising on the target $x_0$ through the latent

variable. All instances of DAs excepting Hudson et al. (2024) tackle unconditional generation, which aligns with the focus of our work. DiffAE (Preechakul et al., 2022) employs an encoder $z = \text{Enc}_\phi(x_0)$ whose output is used at each step of denoising alongside $x_t$ and $t$. InfoDiffusion (Wang et al., 2023a), based on InfoVAE (Zhao et al., 2017), further introduces a probabilistic encoder to maximise MI and align the posterior with a discrete prior of $z$. DiffuseVAE (Pandey et al., 2022) combines VAE and DDPM in a two-stage process, with a DDPM conditioned on the reconstructions of a pre-trained VAE. While these models demonstrate the ability to control the denoising process via a learned latent variable, they share a key limitation in terms of their generative performance. At inference time, they all rely on auxiliary samplers—such as DDIMs (Preechakul et al., 2022; Wang et al., 2023a) or GMMs (Pandey et al., 2022)—to produce valid latent codes, introducing unnecessary overhead.

## 3 DESIGN OF DMZ

A DA with a stochastic encoder $q_\phi(z \mid x_0)$ is primarily trained with a loss that generalises (3):

$$L(\theta) = \mathbb{E}_{x_0 \sim q(x_0),\, \epsilon \sim \mathcal{N}(0,\mathbf{I}),\, t \sim \mathcal{U}(\{1,...,T\}),\, z \sim q_\phi(z|x_0)} \left[ \|\epsilon - \epsilon_\theta(x_t, t, z)\|^2 \right]. \quad (5)$$

Unlike prior approaches that employ additional loss terms to shape the latent space, DMZ directly optimises this objective with respect to the denoiser $\epsilon_\theta$ (now conditioned on $z$) and the encoder $q_\phi$.

In this section, we describe a specific subclass of such diffusion autoencoders (DAs), following a set of judicious design choices, that allow for efficient generative modelling, simultaneously capturing effective latent representations. To begin with, we draw attention to the key distinctions between standard DMs, DMs with a learned forward (noising) process, and a particular type of DA, in Fig. 1.

As can be seen, diffusion models with a learnable forward process (middle) construct a noised observation $x_t$ by additionally incorporating side-information from the observation $x_0$, through a learnable parametric function. The result is that the source for the denoiser $x_t$ can carry additional information to help denoise better to $x_{t-1}$. But this is also a fundamental feature of DAs (bottom)—they incorporate side-information through $z$ into the denoising process by additionally conditioning the denoiser on $z$. In effect, denoising in DAs can be seen as $\{x_t, z\} \mapsto x_{t-1}$, with $x_t$ derived through a standard fixed noising process, just that the information from this $x_t$ and $z$ are not explicitly combined and required to additionally satisfy the constraints of the noising process. Of course, this means that one needs to also be able to sample from the latent $z$ in order to function as a proper generative model; this is what the rest of our design choices seek to address.

**Choice of latent $z$:** In choosing the type of latent variables, we note the importance of discrete latent variables for representation learning. They offer a more interpretable and more space-efficient way to represent data compared to continuous latent variables, and can also capture structured relationships and details, leading to simpler and more effective models (Rolfe, 2017; Vahdat et al., 2018; van den Oord et al., 2017; Razavi et al., 2019; Łukasz Kaiser & Bengio, 2018; Metz et al., 2017). The type (discrete or continuous) and dimensionality of the latent variable forms the basis of the experiments(§4) and ablations (§4.5) we perform.

Diffusion model

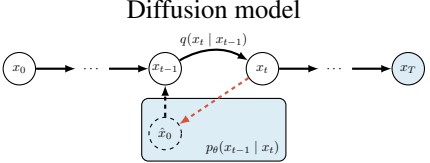

Diffusion model with learned noising

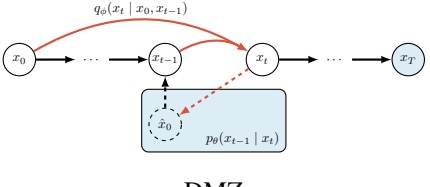

DMZ

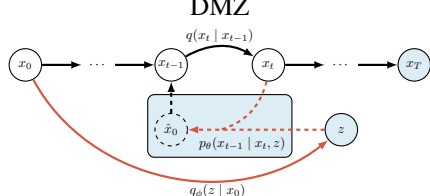

Figure 1: **Top:** Basic diffusion reverses a Markovian noising process from $x_T$ (possibly via predicted $x_0$ at each step). **Middle:** A generalisation where generation reverses a non-Markovian learned noising process, marginalising out unknown $x_0$. **Bottom:** DMZ, where generation conditions on latent $z$. Solid and dashed arrows denote noising and generation respectively. Red arrows denote learned parametric models. Blue objects denote data necessary for generation: initial noise, transition kernel, and $z$.

In the case of discrete $z$, we explore the choice of *binary* latent variables following recent evidence showcasing their effectiveness in the diffusion and reinforcement learning settings Meyer et al. (2024); Wang et al. (2023b). With regard to dimensionality of the latent variable, generally speaking, a lower-dimensional latent is likely to be more interpretable. Conversely, a larger number of dimensions

is capable of capturing more information, helping with use in downstream tasks. An additional consideration comes from how the size of the latent variable affects our ability to model a prior over the latent $z$ to allow sampling at inference time.

**Conditioning on latent $z$:** Another key design decision for effective modelling is the manner in which the denoiser is conditioned on $z$. In the standard (DM) case, the denoiser simply takes the noisy observation $x_t$ along with an encoding of time step $t$. A typical model architecture for such conditioning is shown in Fig. 2 (top), with the time step given as input to each block of a denoising U-Net, where some blocks include self-attention. A natural way to extend this to condition on $z$ would be to include it along with $x_t$ and $t$ as shown in Fig. 2 (middle), as done in some prior DAs (Preechakul et al., 2022; Wang et al., 2023a).

As an alternative, we suggest that $z$ can be more effective when used *only* in modulating attention, and so use *cross-attention* for some blocks with attention layers—with keys $K$ and values $V$ coming from $z$, and queries $Q$ coming from the original inputs to the denoiser. As we will see in the experiments, this turns out to be a useful inductive bias and can have a marked effect on learning effective and useful representations. While our primary experiments use a U-Net backbone, the DMZ framework and its cross-attention conditioning strategy extend naturally to Diffusion Transformers (Peebles & Xie, 2022) (see Appendix A.8).

**Learning with latent $z$:** An interesting feature of DAs is that the latent variable $z$ is in fact largely redundant in terms of captured information from the data. That is, there is no specific pressure for the model to capture *any* information in $z$ given that the standard noising and denoising processes are sufficiently flexible to faithfully model and generate observed data. This lies in direct contrast to typical deep generative models that employ their latent variables as a *bottleneck*, forcing the flow of all information through them.

This feature effectively means that independent constraints placed on $z$, such as regularising it against a typical non-informative prior, as one would in a variational autoencoder, such as the standard normal ($\mathcal{N}(0, I)$), is likely to be quite easily satisfiable and result in the latent becoming non-informative too (see Appendix A.2). This is seen in some prior work (Wang et al., 2023a), where the resulting non-informativity needs further additional regularisation using mutual information with the input. Other approaches avoid this issue by using pretrained probabilistic models with well-defined priors (Pandey et al., 2022).

This points us to models where the prior is flexible enough to capture the data distribution with the generative model, yet simple enough to allow relatively easy definition and capturing of useful latent representations. The choice of latent distribution directly circumvents the apparent redundancy of the latent $z$ in DAs. As we will show with experiments, the binary latents in DMZ offer useful inductive biases: with a small enough latent dimensionality, model learns a uniform posterior distribution—obviating the need for learning the prior either jointly with the model or post-hoc.

Put together, we find that judicious design choices from above allow us to construct a model that (a) does not need auxiliary losses, (b) does not need additional learning of the prior, (c) captures effective representations, and (d) can do all this while being faster to learn than standard diffusion models.

## 4 EXPERIMENTS

We show that DMZ is an efficient and competitive generative model, learns high-quality representations useful for downstream tasks, and extends naturally to a multimodal image-to-image translation framework—all within a unified architecture, for which we provide an ablation study.

All our models are trained following the setup of Nichol & Dhariwal (2021), using their architecture and training hyperparameters. More details can be found in the Appendix and the code[1]. We train until no further improvement in FID scores is observed for $T = 100$ denoising steps. We denote DMZ-$n$ as an instance of DMZ with a latent dimensionality of $|z| = n$. The dimensionality of $z$ was

Unconditional

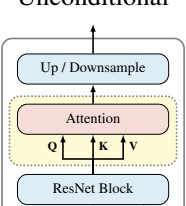

Along with $t$

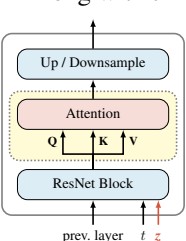

Cross-attention

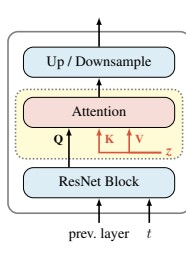

Figure 2: **Top:** Denoiser block with time conditioning and optional attention. **Mid, bottom:** Two conditioning strategies with $z$.

---

[1] https://anonymous.4open.science/r/dmz-47C5

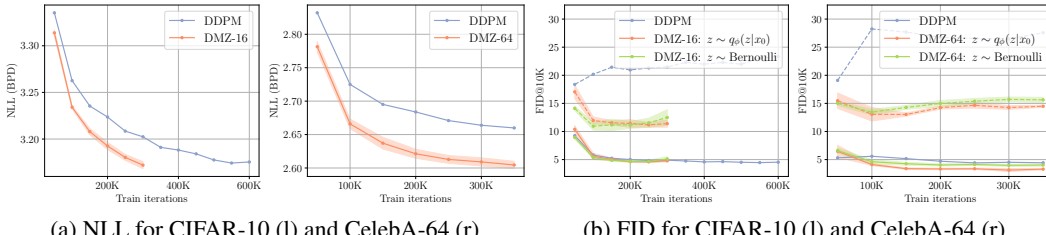

(a) NLL for CIFAR-10 (l) and CelebA-64 (r)      (b) FID for CIFAR-10 (l) and CelebA-64 (r)

Figure 3: Comparison of training curves for DMZ and the baseline DDPM. Dashed lines correspond to results for $T = 10$, while solid lines indicate $T = 100$. For DMZ, we plot mean±std over 5 runs.

deliberately kept small, guided by the number of available labels in each dataset and the requirements of downstream tasks; this choice is further examined in the ablation study. Additional results and samples are provided in the Appendix.

### 4.1 IMPACT OF $z$

**On training and efficiency:** We begin by evaluating the impact of the latent variable $z$ on training efficiency by comparing DMZ to its unconditional counterpart—a standard DDPM. This baseline shares the same architecture and training procedure as DMZ, differing solely in the absence of $z$-specific components. Following prior work, we train models on CIFAR-10 (Krizhevsky, 2009) and CelebA-64 (Liu et al., 2015). We report FID scores calculated using 10K generated samples and the entire dataset (Seitzer, 2020), along with negative log-likelihood (NLL) in bits per dimension.

As shown in Fig. 3, DMZ converges in fewer training iterations and achieves better generation efficiency. Notably, for $T = 10$, it achieves much lower FID scores, demonstrating improved performance when using fewer denoising steps. Moreover, we observe that a lower NLL does not necessarily correspond to better FID scores, highlighting the often-misaligned objectives of likelihood maximisation and perceptual sample quality. We note that our work aligns with prior efforts to improve sampling quality and efficiency in DDPMs, rather than focusing on optimising NLL.

**On the denoising process:** Next, we examine the role of $z$ at different stages of the denoising process (generation). We quantify this by measuring the mutual information (MI) between the learned representations $z \sim q_\phi(z \mid x_0)$ and: (a) noised data $x_t \sim q(x_t \mid x_0, t)$, and (b) denoised generations $x_t \sim p_\theta(x_t \mid x_{t+1}, z)$, $x_T \sim \mathcal{N}(\mathbf{0}, \mathbf{I})$.

For DMZ-16 trained on CIFAR-10, we fix $T = 100$, and compute MI between sampled $z$ and $x_t$ (with $t$ taking 11 evenly spaced values) using MINE (Belghazi et al., 2018)[2]. Fig. 4 illustrates how MI evolves over time.

We find that while $z$ is theoretically redundant when paired with $x_t$ during training, there is non-negligible MI between them, indicating that the network learns to extract meaningful information from $z$. Furthermore, $z$ provides information consistently throughout the denoising process.

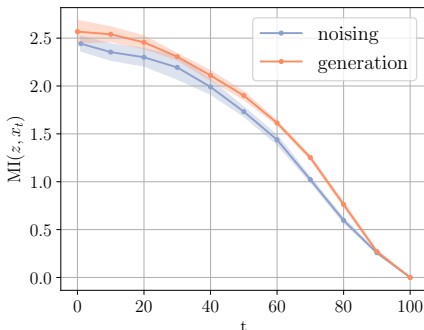

Figure 4: Mutual information between the representations $z$ learned by DMZ-16 on CIFAR-10 and $x_t$ from the noising process (blue) or from the denoising process (orange; starting from $\{x_T, z\}, x_T \sim \mathcal{N}(0, \mathbf{I})$).

### 4.2 GENERATION QUALITY AND EFFICIENCY

We demonstrate that DMZ is an efficient and competitive generative model, outperforming existing diffusion autoencoders (DAs) and diffusion models (DMs) designed exclusively for generation.

Following prior work, we evaluate DMZ-16 and DMZ-64 on CIFAR-10 and CelebA-64 respectively, using FID scores (FID@50K) across various inference step counts ($T = 10, 20, 50, 100$). Results are presented in Table 1, and include scores obtained by an unconditional counterpart to DMZ—a DDPM—which serves as a natural baseline and a reference point, as discussed in §4.1. To ensure fair comparison, we share DMZ scores with 10K samples in Appendix A.9.

---

[2]https://github.com/gtegner/mine-pytorch

First, we observe that DMZ achieves excellent FID scores on both datasets. Thess scores, even with fewer inference steps, highlight efficiency at generation. By comparing DMZ with the DDPM, we again demonstrate, as shown in §4.1, the benefit provided by the addition of $z$.

In the context of DAs, it is important to note that, unlike the baseline DAs which rely on auxiliary samplers for $z$, our model samples $z$ directly from a Bernoulli prior without additional overhead. We see that DiffAE achieves better FID on the CelebA-64 dataset for $T = 10$, which we attribute to its use of DDIM as the base generative model. We combine DMZ with DDIM in Appendix A.3. Overall, we argue that DMZ demonstrates stronger performance and greater simplicity compared to previous DAs.

We find that DMZ achieves performance comparable to state-of-the-art diffusion models (DMs), particularly when accounting for the number of denoising steps required. Additionally, our framework imposes minimal architectural constraints, in contrast to NDFM, which introduces limitations that hinder scalability and flexibility.

### 4.3 REPRESENTATIONS

**Quantitative evaluation:** We assess the quality of the learned representations $z$ following Wang et al. (2023a), by measuring classification accuracy of a logistic classifier trained on the extracted codes. For each dataset, we extract encodings $z$ for the entire dataset, perform a 5-fold split, and report classification performance averaged over the five folds. For CIFAR-10, we report classification accuracy. For CelebA-64, we report the average AUROC over 40 binary attributes to account for class imbalance.

In Table 2, we present how varying the dimensionality of $z$ impacts downstream classification performance. Our results, alongside those of Wang et al. (2023a), show that DMZ, despite applying no explicit constraints or regularisation to $z$ during training, achieves performance equal to or better than InfoDiffusion.

Note that the core idea behind DMZ is to learn representations that provide the denoiser with additional information to guide generation more efficiently. Our goal here is to assess their practical value. While we do not expect state-of-the-art accuracy on downstream tasks, the results in Table 2 demonstrate that the representations are meaningful and compare favorably to our DA baselines. §4.4 further explores their utility through a multimodal framework built upon these representations.

**Qualitative evaluation:** We analyse examples of images generated using representations from DMZ with varying latent dimensionalities $|z|$. For each model, we sample an image from the dataset, $x_0 \sim \mathcal{D}$, and extract its corresponding latent representation. We then generate multiple images by sampling different noise vectors $x_T \sim \mathcal{N}(0, \mathbf{I})$ and generating samples via $p_\theta(x_T, z)$. Fig. 5 presents representative examples illustrating the impact of the size of $z$ on representations. We observe that for smaller $|z|$, less information about the image is retained. Low-level attributes, such as the presence of a smile, are preserved, while higher-level features, such as race, are not consis-

Table 1: FID scores. All DAs except DiffAE use DDPMs (DiffAE only available as DDIM). Models marked $^*$ used 10K samples; all others used 50K. DMZ reports mean and stdev over 5 seeds.

| Model | $T$ | CIFAR-10 | CelebA-64 |
|---|---|---|---|
| DMZ | 10 | 9.78±0.35 | 14.43±0.45 |
| | 20 | 4.79±0.40 | 7.89±0.19 |
| | 50 | 3.04±0.28 | 4.20±0.16 |
| | 100 | 2.79±0.19 | 2.93±0.10 |
| DDPM (Ning et al., 2023) (reproduced) | 10 | 21.25 | 25.72 |
| | 20 | 7.21 | 13.13 |
| | 50 | 3.18 | 5.69 |
| | 100 | 2.56 | 3.34 |
| DiffAE (Preechakul et al., 2022) | 10 | — | 12.92 |
| | 20 | — | 10.18 |
| | 50 | — | 7.05 |
| | 100 | — | 5.30 |
| InfoDiffusion$^*$ (Wang et al., 2023a) | 1000 | 31.5 | 21.2 |
| DiffuseVAE$^*$ (Pandey et al., 2022) | 10 | 34.22 | 25.79 |
| | 25 | 17.36 | 13.89 |
| | 50 | 11.00 | 9.09 |
| | 100 | 8.28 | 7.15 |
| VDM (Kingma et al., 2021) | 1000 | 4.0 | — |
| DiffEnc$^*$ (Nielsen et al., 2024) | 1000 | 14.6 | — |
| NDFM (Bartosh et al., 2024) | 2 | 12.44 | — |
| | 4 | 7.76 | — |
| | 12 | 5.2 | — |
| DDIM (Song et al., 2021a) | 10 | 13.36 | 17.33 |
| | 20 | 6.84 | 13.73 |
| | 50 | 4.67 | 9.17 |
| | 100 | 4.16 | 6.53 |
| | 1000 | 4.04 | 3.51 |

Table 2: Assessment of learned representation quality based on performance in downstream classification tasks. We report averages and standard deviations across 5-fold splits, multiplied by 100.

| Dataset → | CIFAR-10 | | CelebA-64 | |
|---|---|---|---|---|
| Model ↓ | $|z|$ | Acc | $|z|$ | AUROC |
| DiffAE | 32 | 39.5 | 32 | 79.9 |
| InfoDiffusion | 32 | 41.2 | 32 | 84.8 |
| DMZ | 16 | 39.50±0.47 | 32 | 84.85±0.03 |
| | 32 | 42.35±0.56 | 64 | 86.43±0.05 |
| | 64 | 46.45±0.46 | 128 | 87.89±0.02 |
| | 128 | 49.71±0.50 | 256 | 88.48±0.04 |

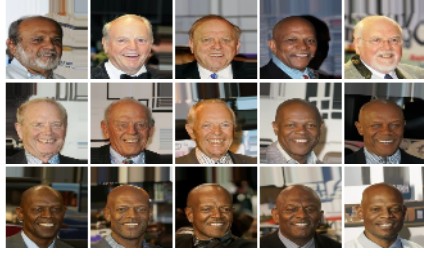

Figure 5: DMZ exemplar generations sharing the same $z$. Rows correspond to $|z| = 64, 128, 256$.

(a) Images generated from interpolations between latent representations of two images using DMZ-256.

(b) Targeted edits using DMZ-256 and CelebA attributes. Edited features from left to right and top to bottom are: *smile*, *open mouth*, *bangs*, *beard*.

Figure 6: Qualitative evaluation on CelebA-64.

tently captured. We attribute this to the fact that high-level
features remain present in the intermediate representations $x_t$ that are passed to the network alongside $z$. As $|z|$ increases, the model is able to encode more information, including higher-level semantic attributes, resulting in generated images that remain consistent across different samples of $x_T$.

Next, we illustrate the properties of the learned representations $z$ through interpolation examples, where transitions between latent vectors lead to gradual changes in attributes such as identity and pose. Specifically, we select pairs of images from the dataset and extract their corresponding encodings, $z_{source}$ and $z_{target}$. We then perform discrete interpolations by sequentially flipping bits in $z_{source}$ to match those in $z_{target}$. Fig. 6a presents examples for DMZ-256 trained on CelebA-64.

Additionally, via the latent variable $z$, we can perform targeted edits on the generated samples—such as altering attributes like hair or facial expression—using classifiers trained on the latent representations. Specifically, we leverage the same classifiers used in the quantitative evaluation and apply edits by moving $z$ along the decision boundary of each classifier. Examples of images generated by using translations of $z$ with $x_T \sim \mathcal{N}(0, \mathbf{I})$, are shown in Fig. 6b.

## 4.4 MULTIMODAL FRAMEWORK

We demonstrate how the DMZ framework can be extended to handle multimodal tasks, specifically focusing on image-to-image translation. Inspired by Denoising Diffusion Bridge Models Zhou et al. (2024), we apply DMZ to the sketch-to-photo translation task – edges2handbags (Isola et al., 2017).

**Model overview:** To effectively reconstruct target $(x_0)$ during translation, the model requires a sufficiently large $|z|$. We train two separate DMZ-512 models: one each for edge images (sketches) and handbags (photos). Training terminates when the mean squared error (MSE) between inputs and samples generated using $z$ shows no further improvement (120K training iterations). The two models learn independent latent spaces—$Z_{sketch}$ for edges and $Z_{photo}$ for handbags. We then learn the mapping between these two latent spaces: $\gamma : Z_{sketch} \rightarrow Z_{photo}$, where $\gamma$ is a mapping function parametrised by a multilayer perceptron (MLP).

**Image translation process:** For sketch-to-photo translation, we follow this pipeline:

(1) **Latent sampling**: We sample a latent variable $z_{sketch} \sim q_\phi(z|x_{sketch})$ from the model trained on sketches, where $x_{sketch}$ is the input sketch image.
(2) **Mapping**: We map sampled sketch latent $z_{sketch}$ into the photo latent space using the learned function $\gamma$, resulting in latent $z_{photo} = \gamma(z_{sketch})$.
(3) **Denoising**: Finally, we use a denoising process to generate the final photo image by sampling $p_\theta(x_T, z_{photo})$ from the photo model, where $x_T \sim \mathcal{N}(0, \mathbf{I})$.

**Results:** Following the evaluation framework of Zhou et al. (2024), we set $T = 40$ and perform sketch-to-photo translations on the training set using our model and baselines. We report FID scores, Inception Scores (IS), LPIPS

Table 3: Evaluation of sketch-to-photo translation task – edges2handbags.

| Model | FID ↓ | IS ↑ | LPIPS ↓ | MSE ↓ |
|---|---|---|---|---|
| Pix2Pix (Isola et al., 2017) | 74.8 | 4.24 | 0.356 | 0.209 |
| DDIB (Su et al., 2023) | 186.84 | 2.04 | 0.869 | 1.05 |
| SDEdit (Meng et al., 2022) | 26.5 | 3.58 | 0.271 | 0.510 |
| Rectified Flow (Liu et al., 2023b) | 25.3 | 2.80 | 0.241 | 0.088 |
| I²SB (Liu et al., 2023a) | 7.43 | 3.40 | 0.244 | 0.191 |
| DDBM (VE) (Zhou et al., 2024) | 2.93 | 3.58 | 0.131 | 0.013 |
| DDBM (VP) (Zhou et al., 2024) | 1.83 | 3.73 | 0.142 | 0.040 |
| DMZ | 3.28 | 3.59 | 0.359 | 0.209 |

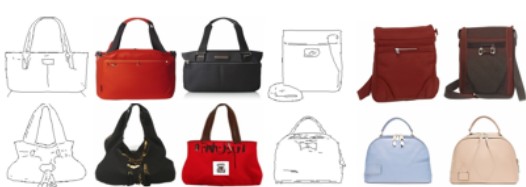

Figure 7: Examples of sketch-to-photo translations using DMZ.

(Zhang et al., 2018), and Mean Squared Error (MSE). Results are shown in Table 3. We achieve competitive performance in comparison to existing approaches, demonstrating effectiveness of the DMZ framework for multimodal image translation. Crucially, DMZ reduces the reliance on expensive joint training across domains. Additionally, our framework supports unconditional generation of images in both domains (photos and sketches), reverse photo-to-sketch translation, and representation learning. Fig. 7 provides examples of the edge-to-handbag translations. Additional examples and discussion are provided in Appendix A.5.

## 4.5 ABLATIONS

We perform an ablation study to analyse the impact of various design choices on the model's abilities and performance. Specifically, we examine the use of a discrete latent space, incorporating the latent variable $z$ into the denoising network, and the size of the latent variable $z$. Each of these choices plays a critical role in shaping the model's overall performance and capabilities.

Additionally, we demonstrate that DMZ can be trained via finetuning a pretrained DDPM. While finetuning leads to some loss in efficiency compared to training from scratch, it offers a faster training process, making it an attractive option when time or memory is a limiting factor. Evaluation details and metrics used are as discussed in previous experiments.

**Discrete $z$:** First, we explore the use of an alternative to discrete $z$, in the form of a Normal prior. We train DMZ-16 on CIFAR-10 using two variants of the prior: discrete (Bernoulli prior) and continuous (Normal prior). In Table 4, we highlight the necessity of using discrete latents. The use of a continuous latent variable makes it infeasible to sample directly from the prior without auxiliary samplers that model the distribution of $z$. This is evidenced by the poor FID scores obtained when sampling directly from the prior, which, in the continuous version, we determine using the mean and standard deviation of the training data. Furthermore, the quality of the learned representations declines, as reflected in the lower accuracy on downstream tasks. We attribute this to the latent space becoming more convoluted, making it more difficult for simple logistic classifiers to perform well. We elaborate more on this ablation in Appendix A.6.

Table 4: Comparison of performance for discrete vs. continuous latents variables. Normal prior is fit over training data parameters. Comparison uses same number of training steps on CIFAR-10

| prior of $z$ | train iter. | NLL (BPD) | Acc | FID@10K prior | $q_\phi(z\|x_0)$ |
|---|---|---|---|---|---|
| Normal | 250K | 3.20 | 34.62 | 34.32 | 5.63 |
| Bernoulli | 250K | 3.18 | 39.50 | 4.79 | 4.56 |

**Conditioning via cross-attention:** We investigate different strategies for incorporating $z$ into the denoising network. The denoising, unconditional UNets consist of ResNet blocks, up/downsampling blocks, and self-attention blocks. These UNets are conditioned on the timestep $t$ by passing $t$ to each ResNet block. A straightforward way to incorporate $z$ is to provide it alongside the timestep $t$ to each ResNet block (Preechakul et al., 2022; Wang et al., 2023a), e.g., by concatenating their embeddings and using the result in place of the standard timestep embedding (Preechakul et al., 2022). However, we find that incorporating $z$ through cross-attention results in better performance. Fig. 2 highlights the architectural differences between the two approaches.

Table 5: Comparison of conditioning methods for $z$ with DMZ-16 on CIFAR-10.

| method | train iter. | NLL (BPD) | Acc | FID@10K Bernoulli | $q_\phi(z\|x_0)$ |
|---|---|---|---|---|---|
| Along with $t$ | 400K | 3.18 | 34.5 | 6.25 | 4.44 |
| Cross-attention | 250K | 3.18 | 39.5 | 4.79 | 4.56 |

To implement this, we replace selected self-attention blocks in the U-Net with cross-attention, enabling better attention over $z$. This improves robustness to $z$ values not seen during training and leads to better learned representations—reflected in both lower FID scores when sampling from the Bernoulli prior and higher accuracy on downstream tasks, as shown in Table 5.

**Small size of $z$:** Here, we discuss how the size of $z$ affects the generative capabilities of DMZ. Clearly, for an autoencoder to accurately reconstruct $x_0$ from $z$, the latent variable $z$ must be sufficiently large. However, when it comes to effective generation, the opposite holds true: a smaller latent space tends to yield better generative performance. Furthermore, a small $|z|$ is sufficient for nearly all use cases. The only exception occurs when reconstruction from $z$ is required, such as in image-to-image translation tasks where Mean Squared Error (MSE) is of concern. Even for image manipulation, a small $|z|$ suffices, as the additional information that a larger $|z|$ could provide is already encoded in the intermediate $x_t$, which is accessible (unlike in image-to-image translation).

We observe that for larger $|z|$, sampling from the Bernoulli prior becomes less effective. To address this, we explore several strategies for sampling $z$ during inference—a critical component for high-quality generation, as evidenced in prior work on DAs. We consider the following three methods:

(1) **Sampling $z$ from data:** For reference, we compute FID scores for $z \sim q_\phi(z \mid x_0)$, where $x_0 \sim \mathcal{D}$ is taken from data. We denote this strategy as $z \sim q_\phi(z|x_0)$.

(2) **Bernoulli Prior:** We sample each latent component independently as $z_i \sim$ Bernoulli($p = 0.5$).

(3) **Autoregressive Prior (PixelSNAIL):** Inspired by prior work on discrete latent models (van den Oord et al., 2017; Razavi et al., 2019), we fit a PixelSNAIL model (Chen et al., 2018) over latent codes to enable sampling. We refer to this sampling method as $z \sim$ PixelSNAIL.

Larger PixelSNAIL models closely match the posterior, or even memorise the dataset, achieving FID scores near those for latents from data. To ensure fair comparison, our models are limited to <600K parameters, based on a grid search that found hyperparameters which provide an optimal balance between performance and model size.

FID scores for all strategies are shown in Table 6. Sampling from PixelSNAIL generally yields better results, particularly in higher-dimensional settings. In lower-dimensional latent spaces, the model better leverages the prior, and sampling directly from it yields strong performance without auxiliary samplers. Therefore, we adopt low-dimensional $z$, optimising for direct sampling.

Table 6: FID score comparison of sampling strategies for DMZ-16 trained on CIFAR-10.

| $T$ | $z \sim \bullet$ | $|z|$ | | | |
|---|---|---|---|---|---|
| | | 16 | 32 | 64 | 128 |
| | $q_\phi(z|x_0)$ | 11.85 | 10.34 | 9.16 | 9.16 |
| 10 | Bernoulli | 11.88 | 10.48 | 15.55 | 22.20 |
| | PixelSNAIL | 11.70 | 10.97 | 11.33 | 14.98 |
| | $q_\phi(z|x_0)$ | 4.56 | 4.96 | 4.61 | 4.46 |
| 100 | Bernoulli | 4.79 | 5.33 | 9.33 | 17.23 |
| | PixelSNAIL | 4.53 | 5.21 | 6.04 | 9.54 |

**Finetuning:** All models presented thus far were trained from scratch. In this section, we investigate the impact of finetuning strategies on the final model performance. Specifically, when time or computational resources are limited, one might opt to finetune a pretrained DM to accelerate training. We explore how this choice affects the capabilities discussed in previous experiments.

We consider three different training strategies: 1) training from scratch, 2) finetuning all parameters, 3) finetuning only the newly added parameters (specifically, those related to the cross-attention mechanism) using a pretrained DDPM. Note that the pretrained DDPM used here is an unconditional DDPM, trained for our previous experiments. Results are presented in Table 7.

Table 7: Comparison of DMZ-64 performance on CelebA-64 for different finetuning strategies.

| Finetuning | train iter. | NLL (BPD) | AUROC | FID@10K | |
|---|---|---|---|---|---|
| | | | | T=10 | T=100 |
| None | 300K | 2.61 | 86.4 | 15.96 | 3.96 |
| All params | 100K | 2.65 | 79.7 | 20.11 | 3.53 |
| New params | 100K | 2.66 | 82.5 | 19.07 | 4.05 |

All models were able to learn effective representations, as demonstrated by their performance on downstream tasks (AUROC). However, the finetuned models did not perform as well in generation tasks with fewer denoising steps, as evidenced by the FID scores for $T = 10$. Overall, all models perform well and are suitable for different use cases, depending on the specific trade-offs between training time, resource requirements, and generation efficiency. In Appendix A.7, we present experiments on finetuning larger-scale DMZ using pretrained models from Hugging Face.

## 5 CONCLUSION

We presented DMZ, a diffusion model inspired by the connection between diffusion autoencoders and diffusion models with a learnable forward process and designed to both learn efficiently and capture effective representations. Through targeted experimentation, we demonstrate that DMZ is capable of generating high-quality samples with fewer denoising steps, while simultaneously learning meaningful representations. Importantly, DMZ achieves these results without the need for additional loss terms, constraints on the latent variable, or auxiliary samplers. Finally, we extend DMZ to a multimodal framework and successfully apply it to an image-to-image translation task, showcasing its versatility and effectiveness. Our findings suggest that the use of additional, input-dependent priors provides a compelling and efficient alternative to traditional diffusion modelling.

**Ethics statement:** This work introduces a generative model that enables controllable image synthesis. By improving efficiency and flexibility in generation, it contributes to advancements in creative AI applications, while raising considerations around responsible use in content creation and editing.

**Reproducibility statement** The implementation and instructions for reproducing the experiments are available at https://anonymous.4open.science/r/dmz-47C5, and will be made public upon publication. Our models are developed using widely adopted libraries, with the most relevant details documented in the Appendix. To ensure robustness, multiple independent runs of the main models were conducted, along with extensive ablation studies that further support the reliability and reproducibility of our findings.

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

# A  APPENDIX

## A.1  DMZ IN RELATION TO PRIOR WORK

In Table 8, we show how DMZ relates to prior work on diffusion autoencoders. Algorithms 1 and 2 outline the training and sampling procedures of DMZ, detailing how the latent variable $z$ is incorporated and highlighting differences from the original DDPM approach Ho et al. (2020).

Table 8: Overview of model features related to training, representations, and evaluation. Diff. loss only – the model is optimised solely using the diffusion loss. Repr. qual. — authors perform quantitative evaluation of representations quality, e.g., via downstream task performance metrics.

| | Training | | Representations | | Evaluation | | | | |
|---|---|---|---|---|---|---|---|---|---|
| | E2E | Diff. loss only | Discrete | No aux sampler | HQ samples | Edits | Repr. qual. | Multi-modal | Fine-tuning |
| DiffAE (Preechakul et al., 2022) | ✓ | ✓ | ✗ | ✗ | ✓ | ✓ | ✓ | ✗ | ✗ |
| InfoDiff (Wang et al., 2023a) | ✓ | ✗ | ✓ | ✗ | ✓ | ✓ | ✓ | ✗ | ✗ |
| DiffuseVAE (Pandey et al., 2022) | ✗ | ✗ | ✗ | ✗ | ✓ | ✓ | ✗ | ✗ | ✗ |
| DMZ | ✓ | ✓ | ✓ | ✓ | ✓ | ✓ | ✓ | ✓ | ✓ |

---

**Algorithm 1:** DMZ training

---

**1** **repeat**
**2**     Sample $x_0 \sim q(x_0)$, $t \sim \mathcal{U}(\{1, \ldots, T\})$, $\epsilon \sim \mathcal{N}(0, \mathbf{I})$
**3**     Compute noisy input $x_t \leftarrow \sqrt{\bar{\alpha}_t} x_0 + \sqrt{1 - \bar{\alpha}_t} \epsilon$
**4**     Extract relaxed code $z$ from $x_0$ via encoder parametrised by $\varphi$
**5**     Take a gradient step on $\nabla_{\theta,\varphi} \|\epsilon - \epsilon_\theta(x_t, t, z)\|^2$
**6** **until** *convergence*;

---

---

**Algorithm 2:** DMZ sampling

---

**1** Sample $\hat{x}_T \sim \mathcal{N}(0, \mathbf{I})$ and $z$ such that $z_i \sim \text{Bernoulli}(p = 0.5)$
**2** **for** $t \leftarrow T$ **to** $1$ **do**
**3**     **if** $t > 1$ **then**
**4**        Sample $v \sim \mathcal{N}(0, \mathbf{I})$
**5**     **else**
**6**        Set $v \leftarrow 0$
**7**     $\hat{x}_{t-1} \leftarrow \frac{1}{\sqrt{\alpha_t}} \left( \hat{x}_t - \frac{1-\alpha_t}{\sqrt{1-\bar{\alpha}_t}} \epsilon_\theta(\hat{x}_t, t, z) \right) + \sigma_t v$
**8** **return** $\hat{x}_0$

---

## A.2  EFFECT OF REGULARISATION AGAINST NON-INFORMATIVE PRIOR

DMZ learns a uniform posterior that aligns with the prior, enabling sampling of $z$ during inference. However, this alignment is not achieved for larger values of $z$. A reasonable solution is to apply regularisation techniques, such as a KL regularisation, to better align the distributions. Table 9 compares the performance of DMZ with and without KL regularisation.

As expected, with regularisation, the models learn to better align the posterior with the prior. However, this leads to less meaningful representations that do not improve the generative performance of the model. While good samples can be generated using the Bernoulli prior, the generative quality is weaker compared to when $|z|$ is restricted to 16 to achieve prior-posterior alignment. Moreover, regularisation negatively impacts the generative performance of the model, even for $|z| = 16$, as it interferes with the denoiser's objective.

Table 9: Comparison of models' performance on CIFAR-10 with and without KL regularisation

| $|z|$ | KL reg. | Acc | FID@10K | |
|---|---|---|---|---|
| | | | Bernoulli | $q_\phi(z|x_0)$ |
| 16 | ✗ | 39.5 | 4.79 | 4.56 |
| 16 | ✓ | 35.8 | 5.05 | 5.23 |
| 64 | ✗ | 45.6 | 9.33 | 4.61 |
| 64 | ✓ | 34.9 | 5.40 | 6.05 |

## A.3 DMZ WITH DDIM SAMPLING

Our baseline DDPM is incompatible with DDIM sampling due to its use of the input perturbation technique (see Section 6.2 in (Ning et al., 2023)) within the DDPM framework[3]. To evaluate the effect of DDIM on DMZ, we trained models without the input perturbation technique. The results in Table 10 indicate that DDIM sampling and DMZ are orthogonal methods, each independently improving generative quality. Note that the DDPM without input perturbation performs worse than our baseline DDPM (see Table 1), which is why we opt to use the latter.

Table 10: Evaluation of the effect of DDIM sampling on DMZ measured via FID@10K.

| Model | DDIM sampling | CIFAR-10 | | | | CelebA-64 | | | |
|---|---|---|---|---|---|---|---|---|---|
| | | 10 | 20 | 50 | 100 | 10 | 20 | 50 | 100 |
| DMZ | ✗ | 15.44 | 8.88 | 5.99 | 5.11 | 28.62 | 13.73 | 6.72 | 4.87 |
| | ✓ | 12.51 | 8.55 | 6.19 | 6.08 | 20.95 | 12.42 | 6.51 | 4.52 |
| DDPM | ✗ | 22.90 | 11.93 | 6.80 | 5.53 | 73.14 | 29.35 | 11.39 | 8.43 |
| | ✓ | 13.17 | 8.21 | 5.54 | 5.29 | 20.34 | 14.91 | 10.31 | 9.41 |

## A.4 INTERPOLATION FORMULATION AND EXAMPLES

Algorithm 3 describes how discrete interpolations between two latent vectors are performed by flipping the disagreeing bits one at a time in random order. For visualisations, we take latent codes equally spaced along the interpolation trajectory. Algorithm 4 details translations across the classifier's decision boundary. Examples of both are shown in Fig. 8 and Fig. 9.

---

**Algorithm 3:** Discrete interpolation between $z_{\text{source}}$ and $z_{\text{target}}$

---

**Input:** $z_{\text{source}}, z_{\text{target}} \in \{0,1\}^n$
**Output:** Interpolation sequence $\{z^{(i)}\}_{i=0}^k$, where $z^{(i)} \in \{0,1\}^n$, $z^{(0)} = z_{\text{source}}$, $z^{(k)} = z_{\text{target}}$
1 Let $\mathcal{I} = \{j \mid z_{\text{source}\,j} \neq z_{\text{target}\,j}\}$;       // Indices where source and target disagree
2 Let $k = |\mathcal{I}|$ and $j_1, \ldots, j_k$ be a random ordering of $\mathcal{I}$
3 $z^{(0)} \leftarrow z_{\text{source}}$
4 **for** $i \leftarrow 1$ **to** $k$ **do**
5 $\quad z^{(i)} \leftarrow z^{(i-1)}$
6 $\quad z_{j_i}^{(i)} \leftarrow 1 - z_{j_i}^{(i)}$ ;                                         // Flip bit
7 **return** $\{z^{(i)}\}_{i=0}^k$

---

**Algorithm 4:** Translations of $z$ across the decision boundary of a binary classifier

---

**Input:** $z \in \{0,1\}^n$, classifier weights $W \in \mathbb{R}^{n \times 2}$ and bias $b \in \mathbb{R}^2$, step sizes with directions $\delta_i \in \mathbb{R}, i = 1, \ldots, k$
**Output:** Interpolation sequence $\{z^{(i)}\}_{i=0}^k$, where $z^{(i)} \in \{0,1\}^n$
1 Let $w_1 \leftarrow W[:,0], w_2 \leftarrow W[:,1]$ ;                            // Class weights
2 $\mathbf{n} \leftarrow w_1 - w_2$ ;                            // Normal vector to decision boundary
3 $\mathbf{v} \leftarrow \frac{\mathbf{n}^\top z + b_1 - b_2}{\|\mathbf{n}\|^2} \cdot \mathbf{n}$ ;                            // Translation vector
4 **for** $i \leftarrow 1$ **to** $k$ **do**
5 $\quad z^{(i)} \leftarrow z + \delta_i \mathbf{v}$ ;                            // Translation of $z$
6 $\quad z^{(i)} \leftarrow \mathbb{I}[z^{(i)} > 0.5]$ ;                            // Optional: binarise vector
7 **return** $\{z^{(i)}\}_{i=0}^k$

---

[3]https://github.com/openai/guided-diffusion

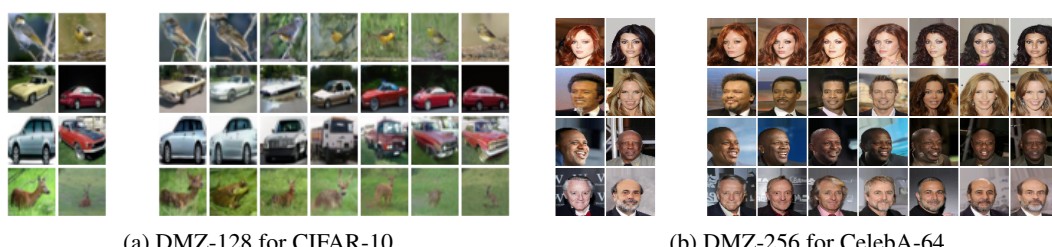

(a) DMZ-128 for CIFAR-10                    (b) DMZ-256 for CelebA-64

Figure 8: Examples of discrete interpolations between codes $z^a$ and $z^b$, where $z^a \sim q_\phi(z|x_0^a)$, $z^b \sim q_\phi(z|x_0^b)$, $x_0^a, x_0^b \sim \mathcal{D}$, and $x_T \sim N(0, \mathbf{I})$. Original images $x_0^a, x_0^b$ are shown in the first two columns.

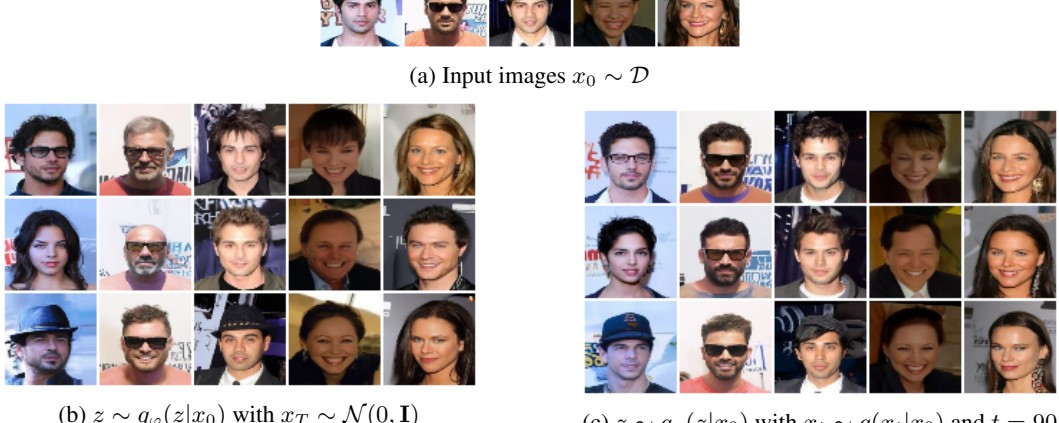

(a) Input images $x_0 \sim \mathcal{D}$

(b) $z \sim q_\varphi(z|x_0)$ with $x_T \sim \mathcal{N}(0, \mathbf{I})$        (c) $z \sim q_\varphi(z|x_0)$ with $x_t \sim q(x_t|x_0)$ and $t = 90$

Figure 9: Examples of classifier-based edits for $T = 100$ on CelebA-64 using DMZ-256. For the first image $x_0$, we change following CelebA attributes: glasses, male, hat; for the second: gray hair, bald, smile; the third: bangs, blond hair, hat; the fourth: bangs, male, earrings; the fifth: blond hair, male, smile.

## A.5    Multimodal DMZ details

To build the multimodal framework—specifically the image-to-image model composed of DMZ modules—we train each component independently and evaluate its performance in isolation. This modular approach allows us to assess the effectiveness of each part before assembling the full model, ensuring that all components function reliably. Below, we describe this process for the multimodal DMZ trained for Edges2Handbags sketch-to-photo task.

**DMZ modules**    We use two instances of DMZ-512: one trained on Edges-64 and the other on Handbags-64. In the sketch-to-photo task, only the model trained on photos is used to generate images, while the model trained on sketches is used to encode their representations. The mean squared error (MSE), defined as $\| x_{\text{photo}} - \widehat{x}_{\text{photo}} \|$, where $\widehat{x}_{\text{photo}} \sim p_\theta(x_{\text{photo}}|z_{\text{photo}})$ and $z_{\text{photo}} \sim q_\varphi(z|x_{\text{photo}})$, serves as an upper bound of the MSE for the sketch-to-photo generation task. We monitor this metric during training and stop once it no longer improves. Additionally, the latent dimensionality $|z| = 512$ was selected based on that MSE performance. Fig. 10 shows the reconstruction error over the course of training.

**Mapping** $\gamma$    We train an MLP to learn a mapping $\gamma : Z_{\text{sketch}} \rightarrow Z_{\text{photo}}$ using latent codes from the DMZ models. To determine the optimal architecture, we experiment

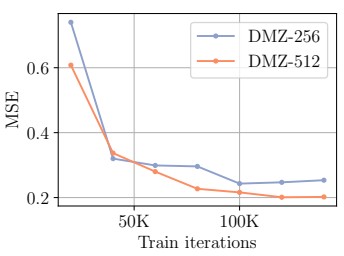

Figure 10: The reconstruction error of DMZ models trained on Handbags-64 measured on 10K images, an upperbound for Edges2Handbags task.

with different numbers of layers $L$ in the MLP and evaluate MSE $\| x_{\text{photo}} - \widehat{x}_{\text{photo}} \|$, where $\widehat{x}_{\text{photo}} \sim p_\theta(x_{\text{photo}} \mid \gamma(z_{\text{sketch}}))$ and $z_{\text{sketch}} = q_\varphi(z \mid x_{\text{sketch}})$. The resulting MSEs for $L = 1, 2, 4, 6, 8$ are 0.26, 0.24, 0.23, 0.22, and 0.23, respectively, leading us to select $L = 6$ as the optimal depth. However, note that a simpler mapping would provide greater interpretability for the framework.

**Additional capabilities of DMZ image-to-image framework**  With our DMZ framework, we can perform reverse image-to-image mapping—generating sketches from photos—as well as unconditional generation of both photos and sketches. Examples are shown in Fig. 11. Note that we use PixelSNAIL for unconditional generation, as the latent size $|z| = 512$ was chosen to optimise reconstruction loss rather than efficient sampling of $z$.

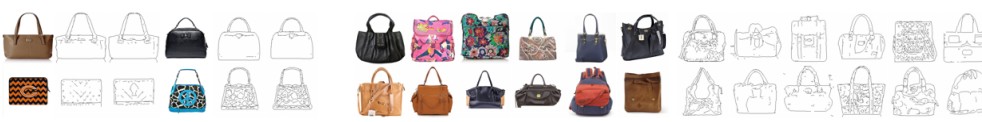

(a) Photo-to-sketch translations          (b) Unconditional generation

Figure 11: Qualitative results showing additional capabilities of the DMZ image-to-image framework.

## A.6 RECONSTRUCTION PERFORMANCE OF DISCRETE VS CONTINUOUS REPRESENTATIONS

A continuous latent variable $z$ generally carries more information about the target $x_0$ than a discrete latent of the same size, enabling better reconstruction. However, this advantage holds primarily when $z$ is the only available information about the target—specifically, at the final timestep $T$. For intermediate noisy inputs $x_t$ with $t < T$, Fig. 12 shows that the reconstruction error becomes comparable between models with discrete and continuous latents.

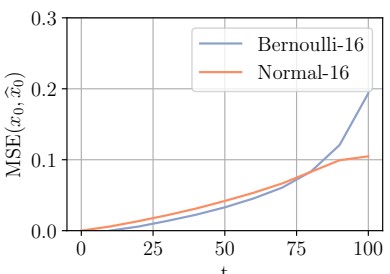

Figure 12: Reconstruction performance comparison between DMZ (Bernoulli-16) and continuous latent $z$ (Normal-16). Mean Squared Error (MSE) is reported on CIFAR-10 for predicting $x_0$ from $x_t$ over $t$ denoising steps, with $T = 100$.

The core idea behind DMZ is to learn latent representations that provide the denoiser with additional, useful information to improve denoising efficiency, while keeping the latent space simple enough to allow easy sampling during inference. Importantly, in DMZ, the latent $z$ is not intended to be sufficient for full image reconstruction on its own. Rather, it serves as a complementary guide for the denoiser. For downstream tasks requiring higher reconstruction fidelity, such as targeted edits or interpolations, one can utilise intermediate states $x_t$. Our main experiments assume $t = T$ to isolate the effect of $z$ alone. Fig. 9 illustrates examples of edits that leverage state $x_t$ to better preserve image identity.

## A.7 FINETUNING HUGGINGFACE MODELS

We have demonstrated that unconditional DDPMs can be effectively finetuned into the DMZ framework, enabling representation learning, conditional generation, and other capabilities. Here, we finetune the publicly available DDPM model trained on CelebA-HQ ($256 \times 256$), available via HuggingFace[4], into DMZ. We train our models for 40K training iterations by finetuning all parameters. Quantitative results are presented in Table 11.

Table 11: Evaluation of DDPM trained on CelebA-HQ and DMZ finetuned from it.

| Model | NLL (BPD) | AUROC | FID@10K | | | |
|---|---|---|---|---|---|---|
| | | | T=10 | T=20 | T=50 | T=100 |
| DDPM | 6.25 | — | 71.43 | 53.55 | 36.86 | 29.81 |
| DMZ-64 | 3.01 | 74.06±0.14 | 39.91 | 28.16 | 19.60 | 15.15 |
| DMZ-256 | 3.00 | 88.53±0.05 | 49.53 | 42.31 | 33.25 | 27.54 |

---

[4]https://huggingface.co/google/ddpm-ema-celebahq-256/

### A.8 DMZ WITH DIFFUSION TRANSFORMERS

To demonstrate the versatility of the DMZ framework, we apply it to the attention-based DiT architecture (Peebles & Xie, 2022). We adapt the model by replacing self-attention layers in every second DiT block with cross-attention layers that attend to a latent $z$.

We train DiT-B/4 models on CIFAR-10 for 200K iterations without specific hyperparameter tuning[5]. As shown in Table 12, DMZ significantly improves generative quality at lower inference steps compared to the baseline DiT, while maintaining an identical NLL of 3.30. Furthermore, the learned representations achieve an unsupervised classification accuracy of $51.52 \pm 0.55\%$, surpassing the U-Net-based results reported in Table 2.

Table 12: Comparison of FID scores (10K samples) on CIFAR-10 between DiT and DiT adapted with DMZ. Reported values are the average $\pm$ standard deviation over 5 runs.

| Model | $T = 10$ | $T = 20$ | $T = 50$ | $T = 100$ |
|---|---|---|---|---|
| DiT-B/4 | $34.11 \pm 0.23$ | $17.59 \pm 0.37$ | $12.83 \pm 0.33$ | $12.19 \pm 0.21$ |
| DMZ | $19.77 \pm 0.22$ | $14.32 \pm 0.09$ | $12.81 \pm 0.12$ | $12.90 \pm 0.08$ |

### A.9 ADDITIONAL SCORES AND SAMPLES

Additional scores are reported in Table 13 and Fig. 13, and additional examples are provided in Figs. 14 to 16.

Table 13: FID scores for DMZ models evaluated with 10K samples. Reported values are the average $\pm$ standard deviation over 5 models.

| Model | Dataset | T=10 | T=20 | T=50 | T=100 |
|---|---|---|---|---|---|
| DMZ-16 | CIFAR-10 | 11.53$\pm$0.90 | 6.76$\pm$0.69 | 4.96$\pm$0.36 | 4.67$\pm$0.24 |
| DMZ-64 | CelebA-64 | 15.70$\pm$0.50 | 8.91$\pm$0.38 | 5.18$\pm$0.27 | 3.84$\pm$0.48 |

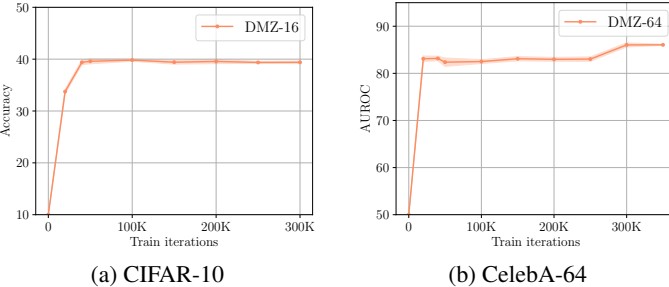

(a) CIFAR-10      (b) CelebA-64

Figure 13: Evolution of representation quality throughout training. High-quality representations emerge early and remain stable throughout.

### A.10 REPRODUCIBILITY DETAILS

We adopt the hyperparameter settings from Ning et al. (2023), which are based on the configurations by Dhariwal & Nichol (2021). The specific values are listed in Table 14. All models are trained using the AdamW optimiser Loshchilov & Hutter (2019) with 16-bit mixed precision training with loss scaling (Micikevicius et al., 2018; Dhariwal & Nichol, 2021), while keeping the model weights, EMA, and optimiser states in 32-bit precision. An EMA decay rate of 0.9999 is used in all experiments, following the setup from Ning et al. (2023).

The encoder used to extract codes $z$ from input images consists of repeated blocks of a convolutional layer, batch normalization, and LeakyReLU activation, followed by a final projection layer. We use 4 blocks for $32 \times 32$ images, 5 blocks for $64 \times 64$, and 7 blocks for $256 \times 256$.

---

[5]https://github.com/facebookresearch/DiT

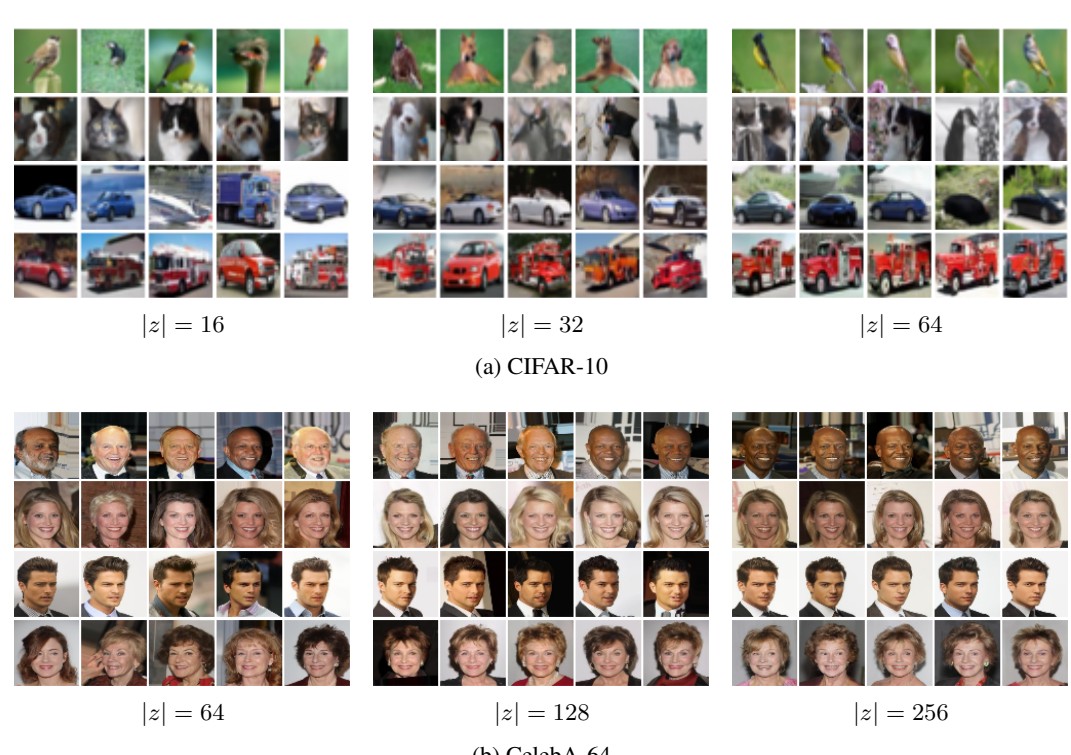

$|z| = 16$ $\qquad\qquad\qquad\qquad$ $|z| = 32$ $\qquad\qquad\qquad\qquad$ $|z| = 64$

(a) CIFAR-10

$|z| = 64$ $\qquad\qquad\qquad\qquad$ $|z| = 128$ $\qquad\qquad\qquad\qquad$ $|z| = 256$

(b) CelebA-64

Figure 14: Comparison of representations learned by DMZ on CIFAR-10 and CelebA-64 for varying latent sizes $|z|$. Images are generated from $z \sim q_\phi(z \mid x_0)$, $x_0 \sim \mathcal{D}$ and five different $x_T \sim \mathcal{N}(0, \mathbf{I})$.

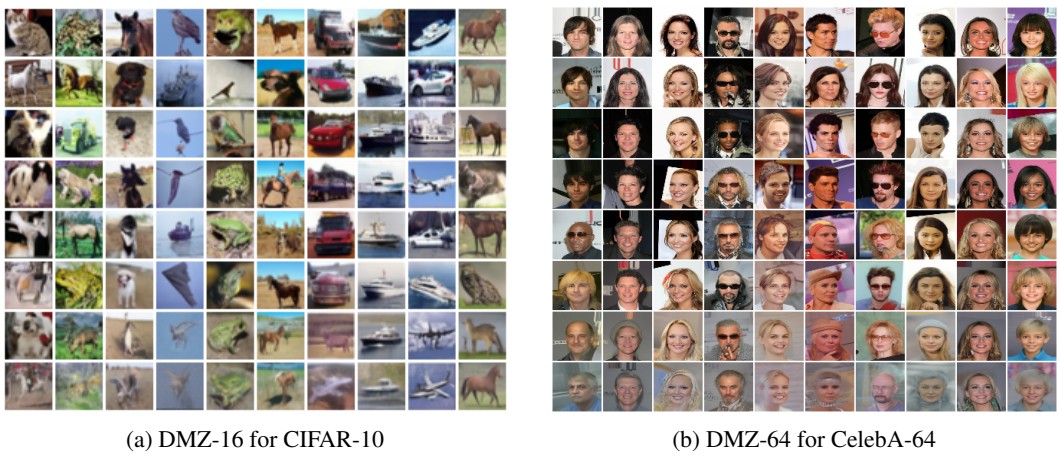

(a) DMZ-16 for CIFAR-10 $\qquad\qquad\qquad\qquad\qquad$ (b) DMZ-64 for CelebA-64

Figure 15: Images generated with varying numbers of denoising steps $T$. Each column shows samples generated from a fixed latent code $z \sim$ Bernoulli. Rows correspond to $T = 1000, 500, 200, 100, 50, 20, 10, 5$ steps, from top to bottom.

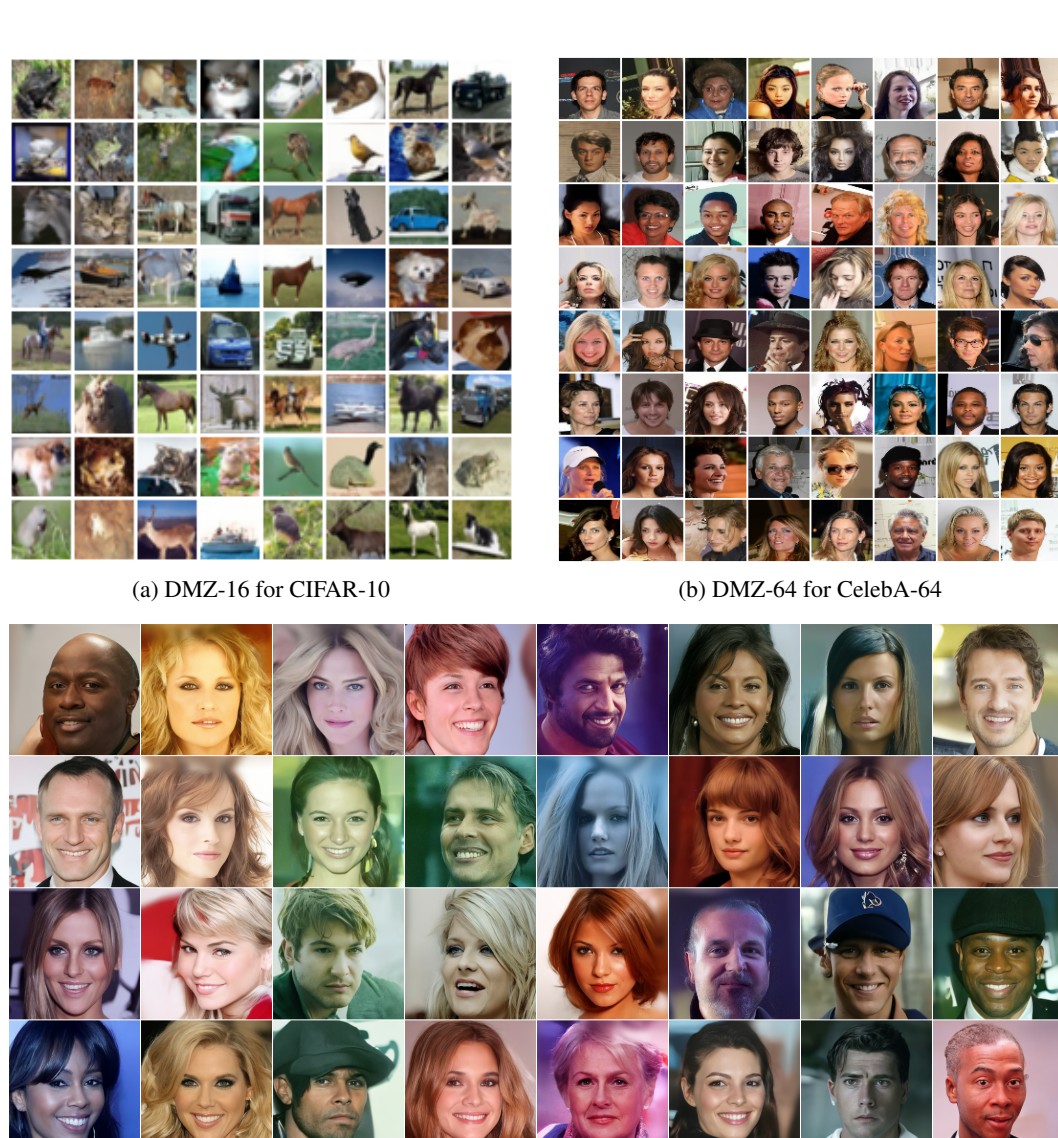

(a) DMZ-16 for CIFAR-10          (b) DMZ-64 for CelebA-64

(c) DMZ-64 for CelebA-HQ

Figure 16: Generated sample images produced using $T = 100$ diffusion steps.

We use PyTorch Paszke et al. (2019), and train all models with Python 3.10 and PyTorch version 2.6. For CIFAR-10, we use a single NVIDIA A40 GPU and train for approximately 2 days. For CelebA-64, we use two A40 GPUs and train for about 10 days. Models trained on Edges2Handbags—handled separately as Edges and Handbags—are also trained using two A40 GPUs, with a training time of around 3 days. Finetuning of CelebA-HQ DDPM (Ho et al., 2020) takes slightly less than 2 days on 4 A40 GPUs. For T=100, sampling a batch of 64 images using a single A40 GPU takes 12s, 264.5s, 79.3s, 79.3s, and 222.5s, for CIFAR-10, CelebA, Edges, Handbags, and CelebA-HQ, respectively.

Our implementation and instructions for reproducing the experiments are available at https://anonymous.4open.science/r/dmz-47C5, and will be made public upon publication.

Table 14: Hyperparameter values based on Ning et al. (2023) for all datasets except CelebA-HQ, where we follow the configuration from Ho et al. (2020) and perform finetuning only.

| | CIFAR-10 $32 \times 32$ | CelebA $64 \times 64$ | Edges $64 \times 64$ | Handbags $64 \times 64$ | CelebA-HQ* $256 \times 256$ |
|---|---|---|---|---|---|
| Size of $z$ | 16/32/64 | 64/128/256 | 512 | 512 | 64/256 |
| Diffusion steps | 1,000 | 1,000 | 1,000 | 1,000 | 1,000 |
| Noise schedule | cosine | cosine | cosine | cosine | linear |
| UNet size | 69M | 409M | 333M | 333M | 142M |
| Encoder size | 0.5M | 1.8M | 3.7M | 3.7M | 29M |
| Channels | 128 | 192 | 192 | 192 | 128 |
| Residual blocks | 3 | 3 | 3 | 3 | 2 |
| Channels multiple | 1, 2, 2, 2 | 1, 2, 3, 4 | 1, 2, 3, 4 | 1, 2, 3, 4 | 1, 1, 2, 2, 4, 4 |
| Heads channels | 32 | 64 | 64 | 64 | 512 |
| Attention resolution | 16, 8 | 32, 16, 8 | 32, 16, 8 | 32, 16, 8 | 16 |
| Cross attention resolution | 16, 8 | 32, 16, 8 | 16 | 16 | 16 |
| Mid-block cross attention | True | True | True | True | True |
| BigGAN up/downsample | True | True | True | True | True |
| Dropout | 0.3 | 0.1 | 0.1 | 0.1 | 0.1 |
| Batch size | 128 | 256 | 256 | 256 | 256 |
| Training iterations | 250K | 300K | 120K | 120K | 40K |
| Training images | 50K | 163K | 139K | 139K | 24K |
| Learning rate | 1e-4 | 1e-4 | 1e-4 | 1e-4 | 1e-4 |
| Learned sigma | True | True | True | True | False |
| Noise schedule | cosine | cosine | cosine | cosine | linear |
| Input perturbation | 0.15 | 0.1 | 0.1 | 0.1 | 0.1 |

