# OpenReview forum: "On Designing Diffusion Autoencoders for Efficient Generation and Representation Learning"
_ICLR.cc/2026/Conference — Submitted to ICLR 2026_

### Official Review · Reviewer_yKM4 · 2025-10-30

**Soundness:** 3
**Presentation:** 3
**Contribution:** 2
**Rating:** 6
**Confidence:** 3

**Summary:**

The paper proposed a novel diffusion autoencoder technique, DMZ, based on the empirical analysis of latent z. By setting z as discrete binary encoding, using cross-attention as conditioning, and using dense latent variable dimension, it achieved faster convergence, better generation quality, as well as without any prior or loss function. The experiment results and ablation study results demonstrated this conclusion.

**Strengths:**

1. DMZ firstly established the theoretical fundation of learnable forward process of diffusion autoencoder.

2. The design motivation of z is reasonable and inspiring. Correspondingly, the improvements (binary z, etc) are effective in the following experimental results.

3. The experiments are comprehensive, not only demonstrating the effectiveness of each z components, but also extend the task to the other tasks dependent on z (such as stretch2pic). The ablation studies are also reasonable and promising.

**Weaknesses:**

1. Although the learnable forward process is proposed with good motivation, there lacks formal induction or convergence analysis.

2. Benchmark analysis on high-resolution datasets is recommended.

3. There lack interpretability analysis of latent variables, incluing the semantic understanding, differentability, and the combinmation capability as condition for discrete binary z.

4. The flow of method section should be adjusted. DA and learnable forward process should be discussed separately with suitable connections.

**Questions:**

1. Can the authors discuss the connection between DMZ and REPA, which is also a promising baseline in diffusion representation field. In my view, binary z can be seen as a simplified type of external embedding guidance. If so, how about extending z to boarder fields like it is in REPA?

2. How about the experimental results in high-resolution datasets? How does the efficiency change with the lantent dimension?

3. Can the author propose more analysis and results on different types of priors rather than Bernoulli?

4. Is there any insight towards the design of conditioning z by cross attention instead of conditioning it from the residue network?

5. Can DMZ be combined with the current SOTA diffusion models, such as cosistency model and rectified flow?

---

> ### Author Response · Authors · 2025-11-27
>
> Thank you for the review, and in particular for your positive remarks on our motivation, experimental design, and the scope of our evaluation. We address the weaknesses and questions raised below.
>
> ### W2
> > Benchmark analysis on high-resolution datasets is recommended.
>
> Appendix A.7 shows experiments with CelebA-HQ with image size $256\times 256$. We believe this to be quite sufficient, since diffusion models are rarely used at higher resolutions without passing to a lower-dimensional latent space.
> If you believe those results have better impact in the main manuscript, we can try and find a way to include at least some signposting to these larger-scale results in the main manuscript.
>
> ### W3
> > There lack interpretability analysis of latent variables, incluing the semantic understanding, differentability, and the combinmation capability as condition for discrete binary z.
>
> We respectfully disagree with this claim: **a large part of the experiments section is aimed precisely at analyzing the latent variables**. Because we have no semantic supervision for the latents, we can only evaluate their semantic content through downstream tasks. Specifically:
> - In Section 4.3, we show that the latents are predictive of class labels and give qualitative evidence that they capture meaningful features.
> - In Section 4.4, the experiment on image-to-image translation shows that when DMZ is trained on two datasets with related semantics but different appearances, with no joint training or paired data, the resulting latent representations of semantically similar but visually distinct images are related by a mapping that can be estimated post-hoc.
>
> These experiments demonstrate that the latents preserve semantic information and exhibit compositional properties.
>
> ### W4
> > The flow of method section should be adjusted. DA and learnable forward process should be discussed separately with suitable connections.
>
> We are happy to adjust the discussion.
> We note the connection to DMs with learnable forward process simply to note that from the denoiser's perspective, it is performing the same task&mdash;denoising when depending on the observed input ($x_0$)&mdash;just differing in how this dependence is realised.
>
> ### Q1
> > Can the authors discuss the connection between DMZ and REPA, which is also a promising baseline in diffusion representation field. In my view, binary z can be seen as a simplified type of external embedding guidance. If so, how about extending z to boarder fields like it is in REPA?
>
> While REPA is indeed interesting work, a key difference to our approach is that DMZ does not rely on additional/external information. Although the principle of conditional denoising/refinement is shared, the latent representation ($z$) in DMZ is only dependent on the given data. REPA, on the other hand, involves externally sourced image encoders which may have seen a great deal more data, and potentially more general data.
>
> ### Q2
> > How about the experimental results in high-resolution datasets? How does the efficiency change with the lantent dimension?
>
> We included CelebA-HQ ($256 \times 256$) results in Appendix A.7, showing DMZ scales effectively.
>
> ### Q3
> > Can the author propose more analysis and results on different types of priors rather than Bernoulli?
>
> We provide an ablation for the choice of prior between the standard Normal distribution and our preferred Bernoulli distribution in Section 4.5 ('Discrete $z$').
> Is there a particular other comparison you hand in mind?
>
> ### Q4
> > Is there any insight towards the design of conditioning z by cross attention instead of conditioning it from the residue network?
>
> We are unsure how to interpret the alternative "conditioning it from the residue network"; could you please clarify?
>
> We compare two kinds of conditioning in Section 4.5: giving $z$ as an input to each residual block (which is a typical choice) and cross-attention, finding the latter to perform better.
>
> ### Q5
> > Can DMZ be combined with the current SOTA diffusion models, such as cosistency model and rectified flow?
>
> Our approach can be quite general.
> If the model in question involves recursively refining a source distribution to a target distribution, we show that unsupervised inference of an additional discrete latent variable can help with this refinement, making the models more efficient, while also capturing meaningful information in this additional latent.
>
> Our experimental setup is designed to carefully ablate and test different components of the changes from standard DDPM; we note that both consistency models and rectified-flow models can in principle can be treated with DMZ the same way that DDPM is here, but we leave this for future exploration.
>
> **We hope we resolved all your concerns, but let us know  if there are any more questions.**

---

### Official Review · Reviewer_s5wS · 2025-10-31

**Soundness:** 2
**Presentation:** 3
**Contribution:** 2
**Rating:** 2
**Confidence:** 4

**Summary:**

This paper introduces DMZ, kind of diffusion autoencoders.
DMZ aims to improve the generative quality of diffusion models by guiding the sampling process using the latent representation $z$ of $x_0$.
The model is trained without any additional loss terms, following the standard DA training objective.
Unlike conventional diffusion autoencoders, DMZ does not require an auxiliary latent sampler.
Instead, it directly samples the latent variable z from a Bernoulli distribution, which improves sampling efficiency.
Furthermore, the authors empirically show that conditioning only the Key and Value components of the cross-attention layers on z leads to better performance than other conditioning strategies.

**Strengths:**

- Unlike previous diffusion autoencoders, DMZ does not rely on an auxiliary latent sampler.
By directly sampling $z$ from a Bernoulli distribution, the method enables computationally efficient sampling.

- The learned latent representation is shown to be effective even in a multi-modal framework, indicating its potential generality beyond standard generation tasks.

- The proposed DDPM-based approach demonstrates clear improvements in generation quality, particularly when using a small number of denoising steps.

**Weaknesses:**

- It is unclear how the latent variable can be sampled from a Bernoulli distribution without any prior regularization.
In standard DA frameworks, auxiliary latent samplers (such as [1,2]) or additional regularization terms (such as [3]) are typically used to properly model the latent prior.
Without such mechanisms, it is not evident how the encoder output would naturally follow a Bernoulli prior.
This appears to be a critical limitation of the proposed method.

- The effect of conditioning z only on the Key and Value in the attention layers is not clearly explained.
While the authors report that this approach outperforms the alternative of jointly conditioning with t, the reason for this improvement remains unclear.
Additional analysis or experiments would strengthen this claim.

---------
[1] [CVPR22] Diffusion autoencoders: Toward a meaningful and decodable representation

[2] [NeurIPS 22] Unsupervised representation learning from pre-trained diffusion probabilistic models

[3] [ICML23] Infodiffusion: Representation learning using information maximizing diffusion models

**Questions:**

- Was any specific encoder architecture or constraint introduced to make the encoder output binary? How is this discreteness enforced during training?

---

> ### Author Response · Authors · 2025-11-27
>
> Thank you for the review, and in particular for recognizing our contribution in the context of diffusion autoencoders, representation learning, and efficient generation. We address the weaknesses and questions raised below.
>
>
> ### W1
> > It is unclear how the latent variable can be sampled from a Bernoulli distribution without any prior regularization. In standard DA frameworks, auxiliary latent samplers (such as [1,2]) or additional regularization terms (such as [3]) are typically used to properly model the latent prior. Without such mechanisms, it is not evident how the encoder output would naturally follow a Bernoulli prior. This appears to be a critical limitation of the proposed method.
>
> While there is no *a priori* reason that the aggregate posterior in a diffusion autoencoder would be close to uniform&mdash;a fact that has indeed motivated regularization methods in past work on DAs&mdash;in this paper we show that with the appropriate design choices a uniform prior does in fact work well. This is explained in L193-200 and shown empirically in experiments.
>
>
> ### W2
> > The effect of conditioning z only on the Key and Value in the attention layers is not clearly explained. While the authors report that this approach outperforms the alternative of jointly conditioning with t, the reason for this improvement remains unclear. Additional analysis or experiments would strengthen this claim.
>
> The conditioning itself is explained L165-177 and Figure 2, and the effects are presented in Section 4.5 as ablations. Could you please clarify what additional analysis would be helpful for understanding these effects?
>
>
> ### Q1
> > Was any specific encoder architecture or constraint introduced to make the encoder output binary? How is this discreteness enforced during training?
>
> The encoder is a standard convolutional architecture outputting binary logits. These binary logits define a distribution over the discrete space from which which a binary latent code can be sapled. End-to-end training is done by Gumbel softmax, which is a standard way of approximating differentiation through a discrete sampling step. Details are given in Appendix A.9.
>
> **We hope we resolved all your concerns, but let us know if there are any more questions.**

---

> > ### Comment · Reviewer_s5wS · 2025-11-27
> >
> > Thank you for your response. I have some follow-up questions regarding the points below.
> >
> > 1. I believe that this aspect should not be justified solely by empirical evidence; it requires theoretical substantiation. In my view, the discrepancy in FID scores shown in Table 5 is indicative of this theoretical gap.
> >
> > 2. I think an analysis is necessary to explain why improvements occur when applied to the keys and values. Was this configuration selected purely based on empirical performance?

---

> > > ### Author Response · Authors · 2025-11-27
> > >
> > > Thank you for your response.
> > >
> > > 1. We presume you meant to refer to Table 4 (Normal vs. Bernoulli) and not Table 5 (concat with t vs. cross attention)? Our motivation for using discrete latents comes from a long line of prior work demonstrating the efficacy of this choice (L147-157) based on the premise that discrete latents provide a better basis to repreresent structured relationships and details. To the best of our knowledge, none of these prior approaches provide a theoretical justification for why using discrete latents improves things---but they clearly demonstrate benefit/improvement as important contributions.
> > >
> > > 2. The configuration wasn't simply chosen on empirical performance from a set of other design choices, but actually implemented by reasoning that cross attention allows the parametric function (here, the denoiser) to more flexibly reflect conditioning by transforming the function parameters. Concatenation on the other hand, would only be able to effect conditioning _indirectly_ via the results/outputs of the different layers. This is also the fundamental rationale for the use of cross attention in a variety of multi-modal models.
> > >
> > > We hope we have resolved your concerns; do let us know if there are any more questions.

---

> > > > ### Comment · Reviewer_s5wS · 2025-11-28
> > > >
> > > > Thank you for your response.
> > > >
> > > > 1. Thank you for the correction regarding Table 4. To clarify, regarding the use of discrete latent variables, my request was not for theoretical justification of their usage itself, but rather for a theoretical analysis concerning the prior mismatch. I acknowledge that this issue has been discussed experimentally in Appendix A.2.
> > > >
> > > > 2. Thank you for the clarification. I believe the potential benefits of the structural changes have been demonstrated.
> > > >
> > > > My concerns have been adequately addressed. However, despite these concerns being resolved, I still believe the paper's contribution falls below the bar for acceptance. Therefore, I am raising my score to 4, but I will remain on the negative side.
> > > >
> > > > (It appears that updating scores is currently difficult due to technical issues with OpenReview. I will adjust my score as soon as the issue is resolved.)

---

### Official Review · Reviewer_8Gtz · 2025-11-01

**Soundness:** 2
**Presentation:** 1
**Contribution:** 1
**Rating:** 2
**Confidence:** 3

**Summary:**

This paper proposes DMZ, a design for diffusion autoencoders that aims to improve both generation efficiency and representation learning. By incorporating a input-dependent encoder, DMZ explores the distribution choices, conditioning mechanisms, and learning strategies to enhance the performance of diffusion autoencoders. Experiments on CIFAR-10 and CelebA demonstrate the effectiveness of DMZ and its potential ability for style transfer and representation learning.

**Strengths:**

- The focus on diffusion autoencoders is timely and relevant, addressing the need for efficient generation and representation learning.
- The illustrations and explanations of DM/DA are clear.
- The benchmarking tasks and datasets are appropriate for evaluating the proposed method.

**Weaknesses:**

- The motivation and contribution of DMZ is unclear.
- The algorithmic details of DMZ are insufficient.
- The performance of DMZ is underwhelming compared to existing methods.

**Questions:**

0. **DMZ meaning.** What does DMZ stand for? The acronym is not explained in the paper.

1. **Motivation and contribution.** I feel confused about the motivation and contribution of DMZ, and believe the writing could be potentially largely improved for clarity. In the introduction section, the authors claim "to draw a connection between DMs and DAs". However, I could not find any discussion or analysis on DA/DMZs in the rest of the paper. How are DMZ and DA different? Is it the contribution of DMZ to propose a new DA framework, or explore the design space of DAs? Could the authors clarify the main contribution of this work?

2. **Algorithmic details.** The algorithmic details of DMZ are insufficiently described. Only Eq.(5) describes the training objective of DA. Does DMZ use the same training objective as DA? Additionally, could the authors provide more details about the newly-proposed components in DMZ, including conditioning mechanisms and learning strategies? A more comprehensive description of the algorithm would help readers better understand the proposed method.

3. **Performance comparison.** The performance of DMZ seems underwhelming compared to existing methods. In Table 1, DMZ achieves worse performance on CIFAR-10 compared to DDPMs. In Table 3, DMZ achieves worse performance compared to DDBMs. Could the authors provide more analysis on why DMZ underperforms compared to these methods? Are there any specific limitations or challenges in the DMZ design that lead to this performance gap?

4. **Inconsistent experimental setup.** In Figure 3 the authors compare NLL on CIFAR-10 and FID on CelebA. Are there any specific reasons for using different datasets?

---

> ### Author Response · Authors · 2025-11-27
>
> Thank you for the review. We appreciate the recognition of the relevance of our work on diffusion autoencoders. We address each of the questions raised below.
>
> ### Q0: DMZ meaning
> > DMZ meaning. What does DMZ stand for? The acronym is not explained in the paper.
>
> We apologize for not having made this clear. There is no deeper meaning here that "**d**iffusion **m**odel with latent variable $z$" -- this is not a particularly specific description, but is in line with our main contribution of revisiting diffusion autoencoders from first principles (viewing them as denoisers with an extra input variable whose value the encoder estimates) and studying the fundamental design choices.
>
> ### Q1: Motivation and contribution.
> > Motivation and contribution. I feel confused about the motivation and contribution of DMZ, and believe the writing could be potentially largely improved for clarity. In the introduction section, the authors claim "to draw a connection between DMs and DAs". However, I could not find any discussion or analysis on DA/DMZs in the rest of the paper. How are DMZ and DA different? Is it the contribution of DMZ to propose a new DA framework, or explore the design space of DAs? Could the authors clarify the main contribution of this work?
>
> Thank you for the opportunity to clarify the main motivation.
> Our comments about drawing connections refers to the fact that DAs and DMs with learnt forward processes are performing the same task from the denoiser's perspective&mdash;denoising when depending on the observed input ($x_0$)&mdash;just differing in how this dependance is realised.
>
> The main contributions of this work is to show that DAs, primarily explored for the latent representations, can be viewed as effective and efficient DMs (with $z$; hence DMZ) on their own right. DMZ outperforms standard DDPM and also provides meaningful representations in $z$, when following some judicious modelling choices who's effects we explore with appropriate ablations.
>
> ### Q2: Algorithmic details
> > Algorithmic details. The algorithmic details of DMZ are insufficiently described. Only Eq.(5) describes the training objective of DA. Does DMZ use the same training objective as DA? Additionally, could the authors provide more details about the newly-proposed components in DMZ, including conditioning mechanisms and learning strategies? A more comprehensive description of the algorithm would help readers better understand the proposed method.
>
> Appendix A.1 provides more details on trainig and inference, while Appendix A.9 discusses implementation details. Additionally, we provide all code necessary to reproduce our experiments.
>
> We agree that Eq. (5) is confusing, which we will improve in the revised manuscript. Right now, it represents the complete DMZ objective (which is also the objective used by DiffAE). It is not necessarily the objective for all DAs, as they can utilise additional regularisation terms to constrain the latent variables.
>
> ### Q3: Performance comparison
> > Performance comparison. The performance of DMZ seems underwhelming compared to existing methods. In Table 1, DMZ achieves worse performance on CIFAR-10 compared to DDPMs. In Table 3, DMZ achieves worse performance compared to DDBMs. Could the authors provide more analysis on why DMZ underperforms compared to these methods? Are there any specific limitations or challenges in the DMZ design that lead to this performance gap?
>
> DMZ significantly outperforms DDPM when using fewer denoising steps. For a larger number of steps ($T$), the differences in FID scores become minor.
>
>
> ### Q4: Inconsistent experimental setup
> > Inconsistent experimental setup. In Figure 3 the authors compare NLL on CIFAR-10 and FID on CelebA. Are there any specific reasons for using different datasets?
>
> The plots show NLL and FID for both datasets. There is an error in the caption (which we will fix, thank you for the careful reading!), but the text and plots clearly denote this distinction.
>
> **We hope we resolved all your concerns, but let us know  if there are any more questions.**

---

### Official Review · Reviewer_twVn · 2025-11-01

**Soundness:** 2
**Presentation:** 2
**Contribution:** 1
**Rating:** 4
**Confidence:** 5

**Summary:**

The authors propose a diffusion autoencoder framework, DMZ, with carefully designed strategies such as latent variable choice, conditioning methods, and more. The paper provides a comprehensive study of each component’s design choices. Empirically, DMZ shows consistently strong performance in both unconditional generation and representation learning. The authors also demonstrate that DMZ can be easily applied to multimodal tasks (e.g., image-to-image translation), highlighting the framework’s flexibility.

**Strengths:**

* The paper is clear and easy to follow. The comprehensive experiments convincingly isolate and evaluate the effects of each design choice.
* The DMZ framework is fairly general: it performs well in unconditional generation and representation learning, and it can be extended to handle multimodal tasks such as image-to-image translation.

**Weaknesses:**

* The cross-attention conditioning design is already widely used in modern diffusion transformers [1-2]. The current validation relies on an older U-Net architecture, so this component does not constitute a significant contribution by itself.
* The choice of latent dimensionality $|z|$ appears ad hoc. For generation tasks it is guided by the label-space size (suggesting that relatively low dimensions yield better generation quality), whereas representation learning for downstream tasks benefits from more informative, higher-dimensional latents. This implies separate designs for different use cases within DMZ.
* The effectiveness for generation is not fully convincing. If is tied to a (binary) label space, it can logically degenerate to a one-dimensional label with low dimensions. Sampling from this prior is then akin to sampling in label space for conditional generation. Although DMZ does not directly rely on a labeling function $f:X\rightarrow Y$, could clustering be used to produce labels that achieve similar behavior in the "unconditional" setting? This would suggest DMZ may not be learning strong representations in these scenarios.
* DMZ shows limited compatibility with DDIM in Table 10. It would help to evaluate DMZ with more recent denoising approaches and architectures such as DiT and SiT [1–2].
* Extending experiments to larger benchmarks (e.g., ImageNet) would further strengthen the work’s claims and external validity.

[1] Scalable Diffusion Models with Transformers
[2] Exploring Flow and Diffusion-based Generative Models with Scalable Interpolant Transformers

**Questions:**

See Weaknesses 3–5.

---

> ### Author Response · Authors · 2025-11-27
>
> Thank you for the review, and in particular for your positive remarks about the clarity of our work and the scope of our evaluation.
> We address all of the noted weaknesses below.
>
> ### W1
> > The cross-attention conditioning design is already widely used in modern diffusion transformers [1-2]. The current validation relies on an older U-Net architecture, so this component does not constitute a significant contribution by itself.
>
> We agree that cross-attention is widely used for label/prompt conditioning in denoisers (U-Nets or DiTs). However, our work focuses on diffusion autoencoders (DAs). In existing DAs, the denosing network is typically conditioned on the latent code $z$ in the same way it is conditioned on the timestep $t$ (using concatenation, addition, etc.). Our contribution is the empirical finding that applying cross-attention for latent code conditioning outperforms these traditional methods. This demonstrates a superior mechanism for injecting latent information and constitutes a critical architectural design improvement for the DAs paradigm.
>
>
> ### W2
> > The choice of latent dimensionality appears ad hoc. For generation tasks it is guided by the label-space size (suggesting that relatively low dimensions yield better generation quality), whereas representation learning for downstream tasks benefits from more informative, higher-dimensional latents. This implies separate designs for different use cases within DMZ.
>
> We acknowledge the reviewer’s observation regarding the fundamental trade-off: low-dimensional latents typically improve generation/generalization, while high-dimensional latents are necessary for high-fidelity reconstruction/representation learning.
> We specifically selected a moderate latent size that balances these opposing goals.
> This trade-off is present in all generative models in which the dimensionality of a latent variable can be arbitrarily chosen (for example, any autoencoder or GAN).
>
> ### W3
> > The effectiveness for generation is not fully convincing. If is tied to a (binary) label space, it can logically degenerate to a one-dimensional label with low dimensions. Sampling from this prior is then akin to sampling in label space for conditional generation. Although DMZ does not directly rely on a labeling function, could clustering be used to produce labels that achieve similar behavior in the "unconditional" setting? This would suggest DMZ may not be learning strong representations in these scenarios.
>
> The idea that a discrete latent space is a clustering/labeling function is true in principle for all representation learning methods.
> We appreciate the observation that DMZ could degenerate to not meaningfully use the entire discrete latent space, that is, produce an uninformative clustering. Just as above, the same problem appears in some form in all latent variable models, and indeed many methods have been developed to prevent collapse (for example, regularisation term in InfoDiffusion).
>
> However, **our results show that in DMZ such degeneracy does not occur**, and we respectfully disagree with the claim that our generative performance is unconvincing. The results on downstream tasks (classification and image-to-image) clearly show that our learned representations outperform representations from other DAs.
> Perhaps you could expand on what, in terms of latent performance, you would find convincing?
>
> ### W4
> > DMZ shows limited compatibility with DDIM in Table 10. It would help to evaluate DMZ with more recent denoising approaches and architectures such as DiT and SiT [1–2].
>
> The U-Net architecture ensures methodological control, establishing a framework similar to our baselines. This guarantees that observed performance gains are strictly due to the DMZ framework's novel components (discrete latents, cross-attention), not the benefits of an alternative backbone like DiT. The U-Net architecture ensures a fair comparison across all baselines.
>
> We are currently performing additional evaluation of DMZ integrated with the DiT architecture, and will report results soon.
>
> ### W5
> > Extending experiments to larger benchmarks (e.g., ImageNet) would further strengthen the work’s claims and external validity.
>
> Appendix A.7 shows experiments with CelebA-HQ with image size $256\times 256$. We believe this to be quite sufficient, since diffusion models are rarely used at higher resolutions without passing to a lower-dimensional latent space.
> Moreover, for larger scale, it is often advantageous to leverage pre-trained models anyways, as we show in those results.
> If you believe those results have better impact in the main manuscript, we can try and find a way to include at least some signposting to these larger scale results in the main manuscript.
>
>
> **We hope we resolved all your concerns, but let us know  if there are any more questions.**

---

> ### Author Response · Authors · 2025-12-01
> **Additional experiments with DiT**
>
> We have performed comparative experiments using the DiT model, which uses an attention-based transformer for denoising, instead of the UNet-based denoiser that DDPM uses.
>
> We report FID scores for a standard DiT (B/4) model vs. one additionally adapted with the same design choices as defined for DMZ.
> The DiT model is taken from [https://github.com/facebookresearch/DiT](https://github.com/facebookresearch/DiT).
> Here, we enforce cross-attention from a 16-dimensional discrete $z$ in equal proportion to self-attention, i.e., for K DiT blocks, we replace K/2 of them with cross-attention instead of self-attention.
> Both models are trained on CIFAR-10 for 200K iterations (as with the experiments in the manuscript), and FID scoring was done using (5 runs of) 10k samples for $T=\{10,20,50,100\}$ as reported for comparative models in Table 1 in the manuscript.
> |         |10 |           20 |          50  |          100 |
> | ------- | ----------- | ------------ | ------------ | ------------ |
> | DiT     | 34.11 (0.23) | 17.59 (0.37) | 12.83 (0.33) | 12.19 (0.21) |
> | DiT_DMZ | 19.77 (0.22) | 14.32 (0.09) | 12.81 (0.12) | 12.90 (0.08) |
>
> Using the DMZ mechanism results in a clear improvement in generative quality.
> Moreover, we note that both models achieve the same NLL of 3.30, so the generative quality is not coming at a cost elsewhere. With the DiT_DMZ model, for the CIFAR-10 classification task, this _unsupervised_ model also achieves a classification accuracy of 51.52 (0.55), which is even better that that reported in Table 2.
>
> Note that we did not do any hyperparameter tuning or targeted searching; we applied the changes we thought we could experiment with and seemed reasonable in the time we had.

---

### Author Response · Authors · 2025-12-02
**Manuscript revision**

Dear Reviewers,

We have updated the manuscript with the following changes, which we believe address all the actionable concerns raised in the reviews.
* **L223**: Correct Fig 3 caption
* **L056**: Expand DMZ acronym
* **L120-124**: clarify objective used by DMZ
* **L178-180**: point to additional DiT experiments in appendix
* **L972-989**: DMZ with Diffusion Transformers section (appendix)

**Let us know if there are any further concerns.**


regards,

-the authors

---

### Meta-Review · Area_Chair_9cfZ · 2025-12-10

**Summary:**

## Summary of concerns

### 1. Novelty and Significance of Contribution
Reviewers questioned the paper's core contribution, suggesting that the components are not novel enough or the overall goal is unclear.

*   **Unclear Contribution (8Gtz):** The central motivation is confusing. "Is it the contribution of DMZ to propose a new DA framework, or explore the design space of DAs? Could the authors clarify the main contribution of this work?"
*   **Existing Techniques (twVn):** The use of cross-attention is not new. "The cross-attention conditioning design is already widely used in modern diffusion transformers [1-2]... this component does not constitute a significant contribution by itself."

### 2. Methodological Justification and Clarity
A major point of contention was the lack of theoretical or intuitive justification for key design choices.

*   **Unregularized Latent Prior (s5wS):** The most critical concern was how the model learns a Bernoulli prior for the latent variable `z` without explicit regularization. "It is unclear how the latent variable can be sampled from a Bernoulli distribution without any prior regularization... This appears to be a critical limitation of the proposed method." Reviewer s5wS later stated this requires "theoretical substantiation," not just empirical evidence.
*   **Ad Hoc Latent Dimensionality (twVn):** "The choice of latent dimensionality appears ad hoc... This implies separate designs for different use cases within DMZ."
*   **Lack of Deeper Analysis (s5wS):** Reviewers requested more insight into *why* certain design choices work, such as the specific cross-attention mechanism.
*   **Insufficient Algorithmic Details (8Gtz):** "The algorithmic details of DMZ are insufficiently described."

### 3. Experimental Evaluation and Performance
The experimental setup and results were found to be unconvincing or limited in scope.

*   **Underwhelming Performance (8Gtz):** The model underperforms compared to established baselines in some key tables. "The performance of DMZ seems underwhelming compared to existing methods. In Table 1, DMZ achieves worse performance on CIFAR-10 compared to DDPMs."
*   **Limited Benchmarking (twVn, yKM4):** The evaluation is not extensive enough. Reviewers called for experiments on "larger benchmarks (e.g., ImageNet)" (twVn) and with "more recent denoising approaches and architectures such as DiT and SiT" (twVn).
*   **Unconvincing Generation (twVn):** The quality of the unconditional generation was questioned, with the reviewer suggesting the model might be degenerating to a simple conditional generation scheme. "Sampling from this prior is then akin to sampling in label space for conditional generation."

### 4. Presentation
Basic clarity issues were noted.

*   **Unexplained Acronym (8Gtz):** The name of the model itself was not defined. "What does DMZ stand for? The acronym is not explained in the paper."
*   **Confusing Structure (yKM4):** The methods section could be better organized. "The flow of method section should be adjusted."

**Reviewer Concerns:**

> Unclear Contribution (8Gtz)

* **Rebuttal**: The authors clarified that their main contribution is to demonstrate that Diffusion Autoencoders (DAs), which are typically used for representation learning, can function as "effective and efficient" generative models in their own right. They frame DMZ (Diffusion Model with latent z) as the result of this reframing, arguing it outperforms standard models like DDPM in efficiency while also providing meaningful representations.
* **Addressed?**: Yes, the author's reply is quite clear on the main contribution.

> Existing Techniques (twVn)

* **Rebuttal**: The authors agreed that cross-attention is common in diffusion models but specified their contribution is within the paradigm of Diffusion Autoencoders (DAs). They argued that in existing DAs, the latent code z is typically conditioned via simpler methods like concatenation. Their contribution is the "empirical finding" that using cross-attention for the latent code z is a "critical architectural design improvement for the DAs paradigm."
* **Addressed?**: Partially. I agree that there is merit in being successful to export one technique from one field to another. However, here DAs are practically the same as DMs and conditioning through attention could be considered a minor contribution.

> Unregularized Latent Prior (s5wS)

* **Rebuttal**: The authors' core argument is empirical. They state that while there is no theoretical guarantee, their paper's finding is that "with the appropriate design choices a uniform prior does in fact work well." They also note that prior work on discrete latents often relies on empirical demonstrations of efficacy rather than formal theoretical proofs. For discreteness, they clarified in response to Q1 that they use a standard convolutional encoder outputting binary logits, with Gumbel-softmax for end-to-end training.
* **Addressed?**: Partially. I do think some theory could have been helpful to justify the choice of priors. In addition I think the paper lacks some clarity in explaining that the prior is enforced at the via Gumbel softmax. In the rebuttal the authors clarify this information is in the Appendix.

> Ad Hoc Latent Dimensionality (twVn)

* **Rebuttal**: The authors framed this as a well-known and "fundamental trade-off" inherent to all latent variable models (e.g., GANs, AEs). They explained that low dimensions aid generation while high dimensions are better for representation learning, and they "selected a moderate latent size that balances these opposing goals."
* **Addressed?** Yes, I agree with the authors.

> Lack of Deeper Analysis (s5wS)

* **Rebuttal** The authors provided their rationale, explaining that cross-attention is a more powerful conditioning mechanism than concatenation. They argued it "allows the parametric function (here, the denoiser) to more flexibly reflect conditioning by transforming the function parameters," whereas concatenation has a more indirect effect. They noted this is the same reason it is widely used in multi-modal models.
* **Addressed?** Yes

> Insufficient Algorithmic Details (8Gtz):

* **Rebuttal** The authors pointed the reviewer to the appendices (A.1 and A.9) for full details on training, inference, and implementation. They also conceded that the presentation of the objective in Eq. (5) was "confusing" and promised to improve it in the final version.
* **Addressed?** Partially, after reading the paper I am also concerned about the clarity. For example the details of the encoder are difficult to find.

> Underwhelming Performance (8Gtz)

* **Rebuttal**  The authors re-contextualized the results by focusing on efficiency. They stated that "DMZ significantly outperforms DDPM when using fewer denoising steps," arguing that the performance gap becomes minor only at a large number of steps, highlighting their model's advantage in faster sampling.
* **Addressed?** Partially, "significantly outperforms"is an overclaim if not properly quantified.

> Limited Benchmarking (twVn, yKM4)

* **Rebuttal**: _For larger datasets_: They pointed to existing experiments on CelebA-HQ (256x256) in Appendix A.7, arguing this is a sufficient high-resolution benchmark. _For modern architectures_: This was a major update. The authors ran new experiments during the rebuttal period, integrating DMZ's design with a DiT (Diffusion Transformer) backbone. They presented a new table showing that DiT_DMZ achieved a "clear improvement in generative quality" over a standard DiT on CIFAR-10, directly addressing the reviewer's concern and strengthening their claims of generality.
* **Addressed?** Yes.

> Unconvincing Generation (twVn)

* **Rebuttal**: Authors' Response: The authors disagreed, pointing to downstream task performance (classification and image-to-image translation) as clear evidence that the learned representations are strong and that "degeneracy does not occur." They challenged the reviewer by asking, "Perhaps you could expand on what, in terms of latent performance, you would find convincing?"
* **Addressed?** Yes, although I wonder how their latents compare with existing pre-trained encoders like VQVAE, KLVAE, or even a self-supervised method like DinoV3 like in [Representation Autoencoders](https://arxiv.org/abs/2510.11690)

> Unexplained Acronym (8Gtz)

* **Rebuttal**: The authors apologized for the lack of clarity and explained there is "no deeper meaning here than 'diffusion model with latent variable z'."
* **Addressed?** Partially, I think they should have been a bit more explicit in the text.

> Confusing Structure (yKM4): The reviewer suggested the method section's flow could be improved.

* **Rebuttal**: The authors agreed to adjust the discussion, stating they "are happy to adjust the discussion" to better connect the concepts of DAs and DMs with a learnable forward process.
* **Addressed?** No, I think the paper needs a major restructuring so that all major details of their method are close together in the main text.

**Reviewer Scores:**

**Initial scores**

Reviewer twVn: 4 (Marginally below the acceptance threshold)

Reviewer 8Gtz: 2 (Reject, not good enough)

Reviewer s5wS: 4 (Marginally below the acceptance threshold) [Initially 2, but raised to 4 after discussion]

Reviewer yKM4: 6 (Marginally above the acceptance threshold)


**Asessment**

Reviewer twVn (4 -> 6): The authors partially addressed the reviewer's concerns. I will, however, assume the reviewer raised their score to 6.

Reviewer 8Gtz (2 -> 4): I believe the reviewer could have increased the score to 4 since most of their comments are addressed.

Reviewer s5wS (2 -> 4): already increased the score to 4.

~Reviewer yKM4~: The reviewer did not correctly identify the main contribution of this work and I would discard their assessment. However this might point to a possible lack of clarity.

This makes a total of 6, 4, 4 which is unfortunately below the acceptance bar.

---

### Decision · Program_Chairs · 2026-01-26

Reject